# A diencephalic circuit in rats for opioid analgesia but not positive reinforcement

Maggie W. Waung[1], Kayla A. Maanum[1], Thomas J. Cirino[1], Joseph R. Driscoll[1], Chris O'Brien [2], Svetlana Bryant[2], Kasra A. Mansourian[1], Marisela Morales [3], David J. Barker[2,3] & Elyssa B. Margolis [1,4 ✉]

Mu opioid receptor (MOR) agonists are potent analgesics, but also cause sedation, respiratory depression, and addiction risk. The epithalamic lateral habenula (LHb) signals aversive states including pain, and here we found that it is a potent site for MOR-agonist analgesia-like responses in rats. Importantly, LHb MOR activation is not reinforcing in the absence of noxious input. The LHb receives excitatory inputs from multiple sites including the ventral tegmental area, lateral hypothalamus, entopeduncular nucleus, and the lateral pre-optic area of the hypothalamus (LPO). Here we report that LHb-projecting glutamatergic LPO neurons are excited by noxious stimulation and are preferentially inhibited by MOR selective agonists. Critically, optogenetic stimulation of LHb-projecting LPO neurons produces an aversive state that is relieved by LHb MOR activation, and optogenetic inhibition of LHb-projecting LPO neurons relieves the aversiveness of ongoing pain.

[1] UCSF Weill Institute for Neurosciences, Department of Neurology, University of California, San Francisco, CA, USA. [2] Department of Psychology, Rutgers University, New Brunswick, NJ, USA. [3] National Institute on Drug Abuse, Neuronal Networks Section, National Institutes of Health, Baltimore, MD, USA. [4] Neuroscience Graduate Program, University of California, San Francisco, CA, USA. ✉email: Elyssa.Margolis@ucsf.edu

Opioids are the most effective pain medications, but the risk of overdose and opioid use disorder limits their clinical utility. Uncoupling the analgesic actions of opioids from those that underlie positive reinforcement is therefore a longstanding goal for pharmacotherapeutic development. Identifying circuits that can drive relief of ongoing pain but not reward in the absence of pain is a critical step towards this goal. The lateral habenula (LHb) may participate in such a circuit, as it is not only activated in a pain setting[1–5] but also by other aversive states including reward omission[6] and animal models of depression[7]. CNS sites involved in pain signaling with reported strong inputs to the LHb include the lateral hypothalamus (LH)[8,9] and anterior cingulate cortex (ACC)[10–12]. Furthermore, efferents from the LHb target pain-responsive regions including the lateral periaqueductal gray, dorsal raphe, parabrachial nucleus, and rostromedial tegmental nucleus[13–15]. While morphine injections that covered a combination of the LHb, medial habenula, and posteromedial thalamus reduce pain-related behavior in an acute pain model[16], whether these effects are due to MOR activation specifically in the LHb is an open question.

Since increased activity in LHb neurons encodes aversive states such as ongoing pain, inhibition of this activity should relieve pain. Pain relief, or the mitigation of the aversive state, can produce negative reinforcement[17]. We previously found that MOR activation can decrease neural activity in the LHb via both postsynaptic hyperpolarization and presynaptic inhibition of glutamate release onto subsets of LHb neurons in naïve animals[18]. Here we investigated the specific LHb input circuit and synaptic mechanism by which MOR activation in the LHb produces pain relief. Among six potential inputs to the LHb, we determined that the glutamatergic innervation from the lateral preoptic area of the hypothalamus (LPO) is both pain-responsive and most strongly inhibited by MOR activation. Importantly, we show that activating MORs in this circuit in the absence of pain does not produce reinforcement, suggesting that targeting this circuit could be a significant advance in pain therapy.

## Results

### MOR activation in the LHb produces pain relief but not positive reinforcement

To examine the behavioral impact of selective MOR activation in the LHb during ongoing pain, we used the spared nerve injury (SNI) model of persistent neuropathic pain and implanted bilateral cannulae above the LHb in Sprague Dawley rats (Fig. 1a). Hypersensitivity induced by SNI consistently persists for well over one month after induction[19]. We evaluated sensitivity to touch by measuring mechanical stimulation thresholds with graded von Frey filaments. Bilateral microinjections of the MOR-selective agonist DAMGO (10 µM; 300 nL/hemisphere) into the LHb increased the average hindpaw withdrawal threshold compared to saline microinjections in the same animals, specifically in injured animals, indicating that DAMGO reduced the mechanical hypersensitivity generated by SNI in male rats (Fig. 1b; Results of statistical analyses including assumption testing reported in Supplemental Table 1). In contrast, intra-LHb DAMGO microinjections in sham-injured male rats had no effect on mechanical withdrawal thresholds compared to saline (Fig. 1b). Consistent with the rat literature[19], we did not observe a significant decrease in withdrawal latency to heat in the Hargreaves test after SNI compared to sham-injured controls, and DAMGO microinjections into the LHb did not alter heat withdrawal latency compared to saline in SNI or sham animals (Supplementary Fig. 1a). Intra-LHb DAMGO reversed mechanical hypersensitivity induced by inflammatory pain in the CFA model (Supplementary Fig. 1c, d).

To evaluate whether MOR activation in the LHb influences the affective experience of pain, we used the place conditioning paradigm in the same group of rats. In a three-chamber apparatus, we paired intra-LHb DAMGO microinjections with one chamber and saline microinjections with the opposite side chamber; the third small chamber served as a neutral area connecting these two. Rats with SNI developed a significant conditioned place preference (CPP) for the LHb-DAMGO-paired chamber, while sham-injured rats did not prefer either chamber following conditioning (Fig. 1d). This indicates LHb-DAMGO is removing an ongoing aversive state producing negative reinforcement rather than producing positive reinforcement via a better than expected experience in the absence of a relevant aversive motivational state.

To rule out the potential confound of off-target effects due to DAMGO entering the CSF space via the nearby third ventricle, we microinjected the same solutions intracerebroventricularly (i.c.v.) in male rats with SNI or sham injury. This manipulation did not influence mechanical withdrawal thresholds compared to saline microinjection in either group (Fig. 1b). Furthermore, similar i.c.v. DAMGO microinjections did not generate a CPP in either SNI or sham animals (Fig. 1d). Therefore, we conclude that the behavioral effects of our DAMGO microinjections were due to actions specifically in the LHb.

We also investigated whether LHb MOR activation had the same effects on allodynia and affective pain in female rats. Using the same microinjection parameters as in male rats, female rats with SNI showed a trend towards reduced mechanical allodynia following DAMGO microinjections into the LHb compared to saline (Fig. 1c). Female rats with SNI also showed only a trend towards a preference for the DAMGO-paired chamber (Fig. 1e). As in males, females with SNI displayed no difference in heat withdrawal latency between DAMGO and saline microinjections (Supplementary Fig. 1b). Because female rats may be less sensitive to the analgesic effect of opioids[20,21], we tested a 10-fold higher concentration of intra-LHb DAMGO (100 µM) in a separate cohort of females. This dose was chosen based on studies demonstrating a three-fold reduction in i.c.v. DAMGO antinociceptive efficacy for females in the tail-flick test[22]. The higher DAMGO dose induced a significant CPP in female rats with SNI, though it still did not reverse mechanical allodynia (Fig. 1c, e). Thus, we conclude that LHb MOR activation can reverse the affective experience of pain in both male and female rats.

### MOR synaptic function persists in LHb neurons in animals with chronic pain

In various CNS regions, chronic pain induces changes in MOR expression and function, including receptor downregulation[23–25]. We previously reported that in naïve male rats MOR activation inhibits glutamate release onto a subset of LHb neurons and also hyperpolarizes ~30% of LHb neurons[18]. Here we tested whether these MOR effects are altered in a persistent pain state. To evaluate postsynaptic MOR function we performed whole-cell voltage-clamp recordings of LHb neurons from acute brain slices from male rats with SNI and measured DAMGO induced changes to holding current (Fig. 2a, b). These responses did not differ from our observations in naïve rats in mean response magnitude (Fig. 2b) or frequency of responsive neurons (two-tailed Fisher exact test $p = 1$). We also tested MOR inhibition of glutamatergic electrically evoked excitatory postsynaptic currents (EPSCs) in LHb neurons from animals with SNI. These responses were also consistent in magnitude and frequency (two-tailed Fisher exact test $p = 1$) with observations from naïve rats (Fig. 2c, d). We conclude that these actions of MOR on LHb cell bodies and glutamatergic terminals do not change in animals with ongoing pain.

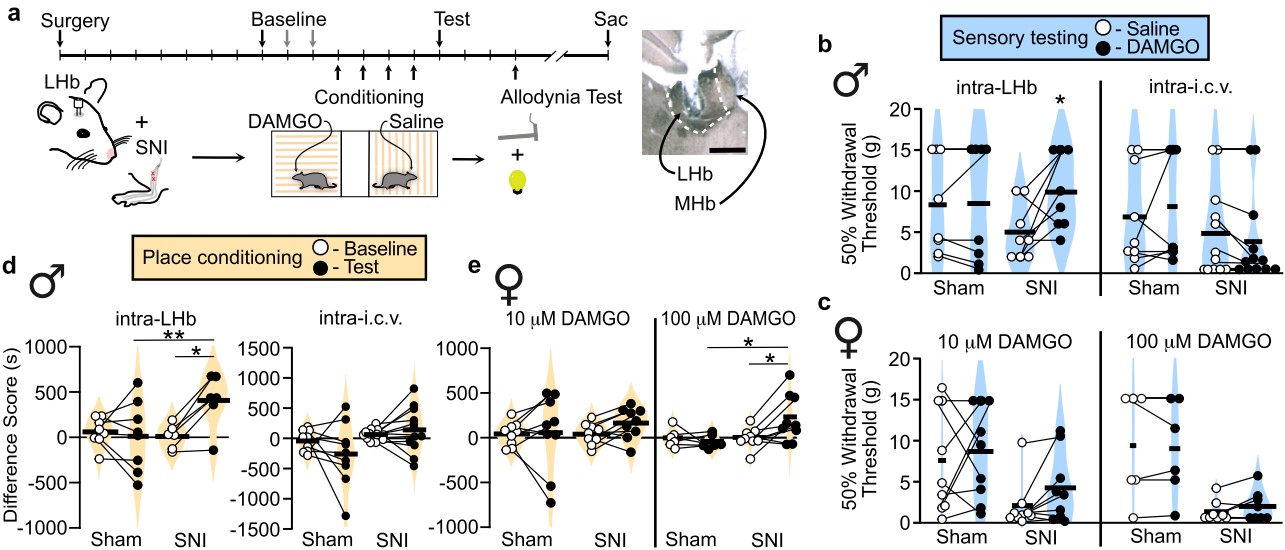

**Fig. 1 MOR activation in the LHb relieves allodynia and generates negative reinforcement in rats with neuropathic pain. a** Timeline of experiments. Right inset, example injection site centered in LHb. Scale bar = 500 μm. **b** (Left) Violin plots of 50% mechanical withdrawal thresholds following intra-LHb microinjections of saline or 10 μM DAMGO in sham ($n = 8$) or SNI ($n = 8$) male rats. Wilcoxon-signed rank test, Sham: $V = 1.5$, $p = 1$. SNI: $V = 2$, $p = 0.050$. (Right) 50% mechanical withdrawal thresholds following saline or 10 μM DAMGO microinjections into the 3rd ventricle (i.c.v.), in SNI ($n = 12$) or sham ($n = 9$) male rats. Wilcoxon signed rank test, Sham: $V = 7$, $p = 0.53$. SNI: $V = 31$, $p = 0.080$. **c** (Left) 50% mechanical withdrawal thresholds following intra-LHb microinjections of saline or 10 μM DAMGO to sham ($n = 9$) or SNI ($n = 9$) female rats. Bonferroni corrected paired $t$-test, $p = 0.703$. Wilcoxon signed rank test, $V = 7$, $p = 0.074$. (Right) 50% mechanical withdrawal thresholds following saline or 100 μM DAMGO intra-LHb microinjections of sham ($n = 6$) or SNI ($n = 8$) females. Wilcoxon signed rank test, Sham: $V = 5$, $p = 1$. SNI: $V = 6$, $p = 0.40$. **d** Difference scores in sham ($n = 8$) or SNI ($n = 6$) males before and after conditioning with intra-LHb microinjections of 10 μM DAMGO vs saline. Two-way mixed ANOVA, significant interaction between SNI/sham and baseline/test $F(1,13) = 15.93$, $p = 0.002$; post hoc effect group on test day adjusted $p = 0.01$; paired $t$-tests, sham adjusted $p = 1$; SNI adjusted $p = 0.016$. (Right) Difference scores in sham ($n = 9$) and SNI ($n = 12$) males before and after conditioning with intra-i.c.v. microinjections of 10 μM DAMGO vs saline: Two-way mixed ANOVA, $F(1,19) = 2.239$, $p = 0.15$. **e** (Left) Difference scores in sham ($n = 9$) and SNI ($n = 9$) females before and after conditioning with intra-LHb injections of 10 μM DAMGO vs saline. Sham: Bonferroni corrected $t$-test $p = 0.703$. SNI: Wilcoxon-signed rank test, $V = 7$, $p = 0.07$. (Right) Difference scores in sham ($n = 6$) and SNI ($n = 9$) females before and after conditioning with intra-LHb injections of 100 μM DAMGO vs saline. Two-way mixed ANOVA, significant interaction between SNI/sham and baseline/test $F(1,13) = 6.234$, $p = 0.027$; post hoc effect group on test day adjusted $p = 0.07$; paired $t$-tests, sham adjusted $p = 0.48$; SNI adjusted $p = 0.035$. Data was collected across a minimum of two cohorts of animals. Means represented by horizontal lines. Tests are two-tailed. *$p \leq 0.05$, **$p \leq 0.01$.

Increased activity in LHb neurons as well as increased glutamatergic synaptic strength onto LHb neurons are associated with aversive behavioral states[7,26,27]. To evaluate this in LHb neurons from animals with SNI, we evaluated the paired pulse ratio of the electrically evoked EPSCs, a measure of probability of release. There was no difference in paired pulse ratio between groups (Fig. 2e). We next compared the frequency and magnitude of spontaneous glutamatergic EPSCs (sEPSCs) in animals with SNI to those in naïve animals. Mean sEPSC frequency and amplitude were also similar in LHb neurons from SNI and naïve animals (Fig. 2f, g). Together, these observations suggest that painful injury does not induce glutamatergic synaptic plasticity in the LHb.

**The LHb receives functional synaptic input from the LPO, LH, VTA, VP, and EPN, but not the ACC.** LHb neuron firing activity increases with acute noxious stimulation, and an increase in ongoing firing frequency is observed during aversive behavioral states[2,3,7]. In ongoing pain, since we did not observe evidence for changes in glutamatergic synaptic strength, such increases in firing may be driven by greater activity in the glutamatergic axons innervating LHb neurons. Therefore, we hypothesized that the intra-LHb DAMGO-induced behavioral effects that we observed in injured animals were due to MOR inhibition of glutamatergic axon terminals, thus decreasing the aversive excitatory drive onto the LHb neurons. Since MOR activation only inhibits

glutamatergic inputs onto a subset of LHb neurons[18], MORs might be preferentially expressed on specific afferent inputs. Prior work characterizing direct functional synaptic connections to the LHb is limited to mice[28], therefore we sought to confirm these functional connections in the rat. We investigated inputs from the entopeduncular nucleus (EPN), lateral preoptic area of the hypothalamus (LPO), and ventral tegmental area (VTA) because stimulating glutamatergic LHb inputs from these sources has been shown to be aversive[26,29,30]. We also investigated inputs from the LH and ACC because they are strongly implicated in pain processing[31], and the VP because stimulating glutamatergic VP neurons increases the firing rate of LHb neurons[32]. We injected AAV2-hSyn-hChR2(H134R)-mCherry into one of these six regions in order to express channelrhodopsin (ChR2) in these different input populations (Fig. 3a). We then made whole cell recordings in LHb neurons and measured light-evoked synaptic inputs while blind to injection site. ChR2 was activated by an LED ($\lambda = 473$ nm) coupled to an optic fiber placed ~100 μm from the recorded cell. Post synaptic currents (PSCs) were measured in response to paired light pulses (1 or 5 ms, 50 ms inter stimulus interval) at holding potentials of −60 mV and −40 mV to probe for EPSCs and GABA$_A$R mediated inhibitory PSCs (IPSCs) in each cell, respectively. Roughly similar proportions of LHb neurons received synaptic input from each of these targets, with the exception of the ACC where we did not detect any fast PSC connections (Fig. 3b, topographical distribution of connected neurons in Supplementary Fig. 2). The absence of a functional

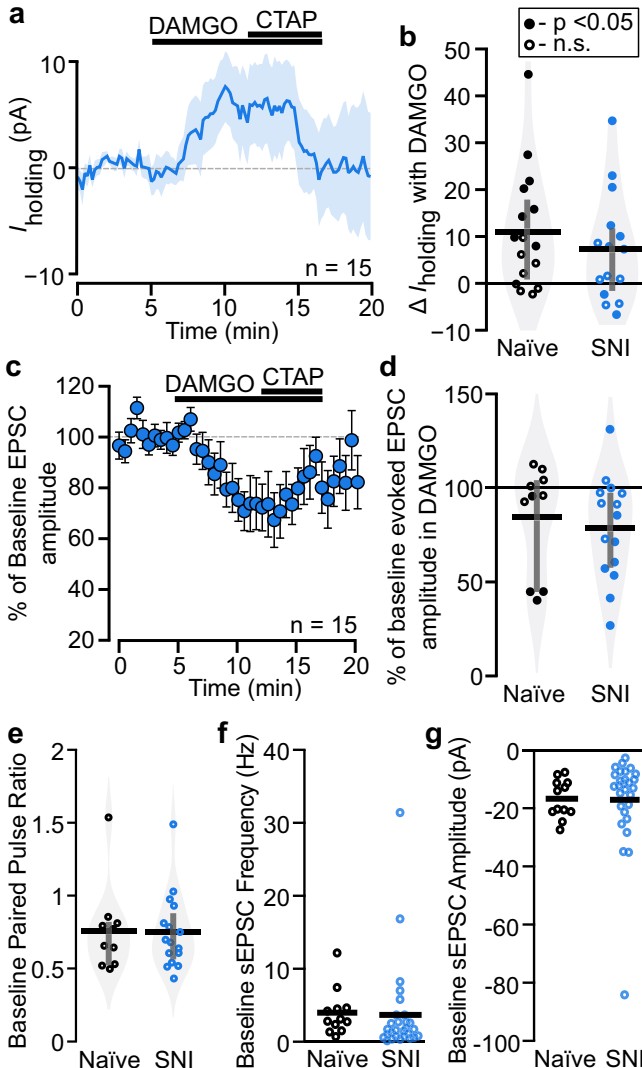

**Fig. 2 Ongoing pain does not alter MOR agonist synaptic effects or probability of glutamate release in the LHb.** Responses to the MOR agonist DAMGO were recorded in voltage clamp, $V_m = -60$ mV. **a** In animals with SNI, a subset of neurons responded to bath application of the MOR agonist DAMGO (500 nM) with a time locked outward current that was reversed with the MOR selective antagonist CTAP (500 nM). Line indicates mean, shading indicates SEM. $n = 15$ neurons from 4 rats. **b** Postsynaptic holding current changes in response to bath application of DAMGO were not different between neurons from naïve and SNI rats: Unpaired $t$-test df = 29, $t = 0.87$, $p = 0.39$. Filled circles indicate cells statistically determined to be "responsive" to DAMGO with within cell unpaired $t$ tests. Gray bars indicate 25th and 75th percentiles. Naïve data: $n = 17$ neurons from 7 rats. **c** In rats with SNI, DAMGO (500 nM) induced a time-locked inhibition of electrically evoked glutamatergic EPSCs, and CTAP (500 nM) partially reversed the effect. Mean ± SEM. $n = 15$ neurons from 4 rats. **d** This DAMGO inhibition of electrically evoked EPSC amplitude was not different between LHb neurons from naïve and SNI rats: Unpaired $t$-test df = 23, $t = 0.47$, $p = 0.64$. Naïve data: $n = 10$ neurons from 4 rats. Baseline probability of release at glutamatergic synapses was similar between LHb neurons in naïve and SNI rats as measured by (**e**) paired pulse ratio (P2/P1) of electrically stimulated EPSCs (50 ms interstimulus interval): Unpaired $t$-test, df = 24, $t = 0.102$, $p = 0.92$; (**f**) baseline spontaneous EPSC frequency: Unpaired $t$-test unequal variances, df = 37, $t = -0.17$, $p = 0.87$; and (**g**) spontaneous EPSC amplitude: Unpaired $t$-test unequal variances, df = 38, $t = -0.09$, $p = 0.93$. Data from naïve rats previously published in[18]. Tests are two-tailed.

synaptic input from the ACC to the LHb was surprising, as both anterograde[10,12] and retrograde tracers[11] have previously indicated modest inputs. Moreover, the ACC is extensively implicated in behavioral responses to pain and MOR-agonist-induced pain relief[33,34]. As a secondary measure of the strength of the innervation, we performed a systematic evaluation of the potential connection using ChR2 as an anterograde tracer, making large injections of AAV2-hSyn-hChR2(H134R)-mCherry throughout the anteroposterior range of the ACC (Supplementary Fig. 3a). This tracing revealed extensive innervation of the nearby mediodorsal thalamus (MDL), but minimal stereologically-quantified fiber labeling in the LHb (Supplementary Fig. 3b, c). Therefore, while we cannot completely rule out a functional input from the ACC to the LHb, any innervation is extremely small compared to the other sources of input to the LHb investigated here.

For each brain region from which fast synaptic PSCs were detected in the LHb, both glutamate and GABA inputs were observed in varying proportions (Fig. 3c). Interestingly, while for each input there were individual LHb neurons that received both glutamate and GABA synaptic connections, for each input more than half of the connected LHb neurons received just one type of fast PSC. The VTA was the only input where more LHb neurons received GABAergic synaptic connections than received glutamatergic synaptic connections. Among observed synaptic connections, a wide range of EPSC and IPSC amplitudes were observed for most of the inputs, except for the glutamatergic inputs from the VTA that were consistently small (Supplementary Fig. 4a, c). The delay to light-evoked EPSC onset also varied across input source, with LH inputs having the shortest mean latency (Supplementary Fig. 4b).

The nature of local LHb neural connections will also impact the circuit's response to MOR activation. There are strong local glutamatergic connections within the LHb[18,35], but there is recent evidence both for[36,37] and against[38,39] the existence of local GABA interneurons. As we only observed somatodendritic MOR responses in a subset of LHb neurons, evidence for GABAergic interneurons in the LHb would impact our model of how MOR activation modulates LHb neural activity, if MOR is preferentially expressed in such interneurons. In the rat, a small number of GAD1 positive neurons are present in the lateral LHb, though these neurons do not co-express vesicular GABA transporter[38], the protein required for loading GABA into synaptic vesicles. In order to detect functional local GABAergic connections within the rat LHb, we injected AAV2-hSyn-hChR2(H134R)-mCherry into the LHb and recorded from LHb neurons. Because some recorded neurons expressed ChR2, we measured light responses before and after application of receptor antagonists in order to isolate the synaptically driven response from the ChR2 mediated currents. Under these conditions we did not observe any light activated local IPSCs in LHb neurons (Supplementary Fig. 5). As expected, many neurons received local glutamatergic inputs (Supplementary Fig. 5). We conclude that there is very limited or no local GABAergic interneuron connectivity in the LHb of adult rats.

**MOR activation most strongly inhibits LPO inputs to the LHb.** We tested for functional MOR modulation of the light-evoked glutamatergic inputs to LHb neurons from each of the regions characterized above. DAMGO induced the strongest and most consistent inhibitions in the terminals arising from LPO neurons (Fig. 3d, e). On average the inhibition was greater in these LPO inputs than the MOR impact on ESPCs observed from the LH, VTA, VP, or EPN (Fig. 3d). We also tested whether MOR inhibits LPO glutamatergic inputs to the LHb independent of sex; the mean inhibition of glutamate release from LPO terminals to LHb

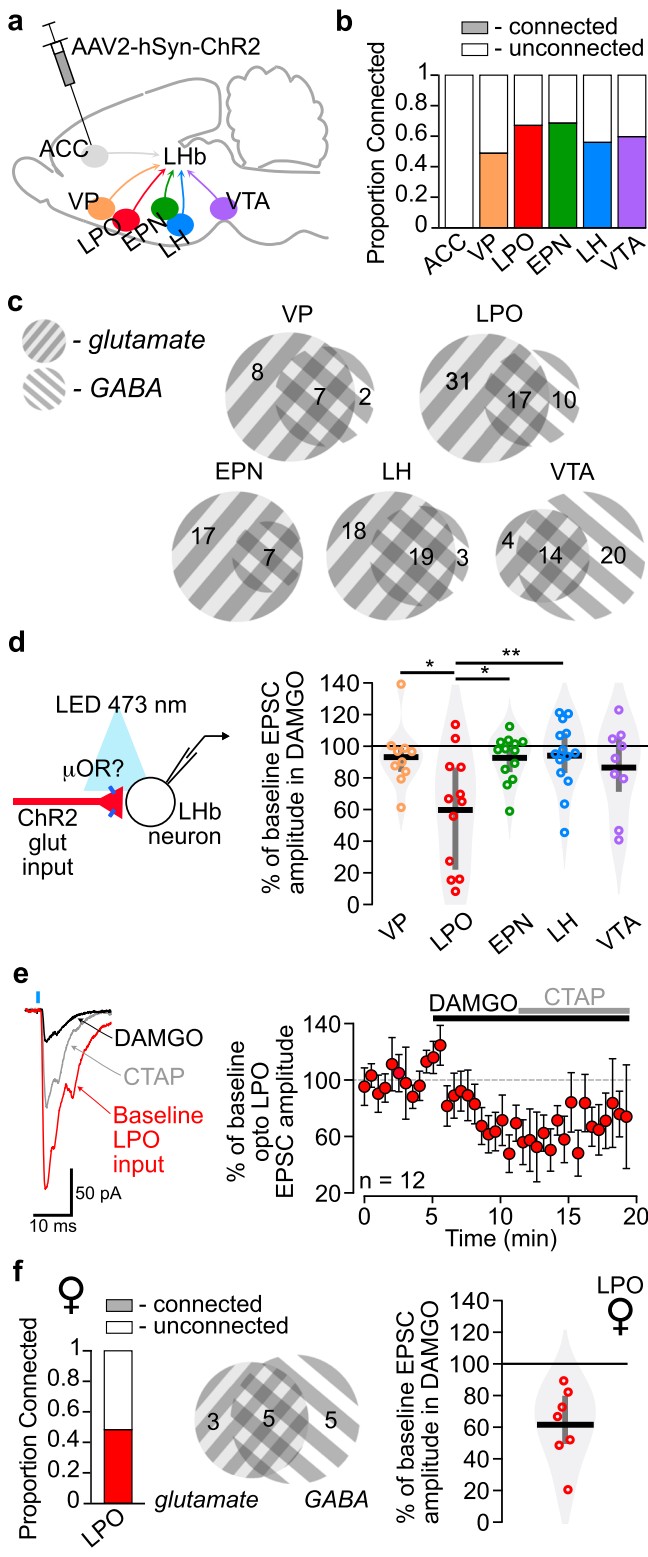

**Fig. 3 Glutamatergic inputs to the LHb from the LPO are inhibited by MOR activation. a** Schematic diagram of AAV2-hSyn-hChR2(H134R)-mCherry injection targets which project to the LHb. **b** Proportions of LHb neurons receiving functional synaptic connections from six input regions measured with optogenetic stimulation of terminals expressing ChR2. **c** Among connected neurons ($n = 17$–$58$), venn diagrams depict the numbers of LHb cells receiving optically stimulated glutamatergic and/or GABAergic synaptic input. **d** (Left) Experiment design: DAMGO (500 nM) was applied during optical synaptic stimulation experiments to probe for functional MOR presynaptic control of glutamate release from specific inputs. (Right) Change from baseline amplitude of optically-evoked EPSCs during DAMGO application. EPSCs from LPO terminal stimulation were most strongly inhibited by MOR activation: One-way ANOVA, df = 4, F = 4.11, $p = 0.0057$ followed by Tukey HSD pairwise comparisons. Data are expressed as mean with 25th and 75th percentiles **e**, (Left) Representative traces of optically-evoked EPSCs in an LHb neuron from a rat with ChR2 expressed in inputs from the LPO. DAMGO decreased the amplitude of light-evoked EPSC, which was partially reversed by the selective MOR antagonist CTAP (500 nM). Rise time was <2 ms, indicative of a monosynaptic response. (Right) Summary time course of the DAMGO inhibition across all optically-evoked LPO-LHb EPSCs ($n = 12$ cells/9 rats), indicated as the mean (circle) ± SEM. **f** LPO inputs to the LHb in female rats (left) had a similar overall connectivity rate but (middle) more GABAergic connections than glutamate connections. (Right) DAMGO inhibited light-evoked glutamatergic synaptic inputs from the LPO to the LHb in female rats as well. Means represented by horizontal lines with 25th and 75th percentiles. Tests are two-tailed. *$p < 0.05$, **$p < 0.01$.

monosynaptic connections, we expressed ChR2 in LPO neurons and recorded in the LHb; in neurons with light-evoked EPSC responses, we applied tetrodotoxin (TTX, 500 nM) and 4-aminopyridine (4 AP; 10 μM). In 8 of 8 tested neurons the light-evoked EPSCs persisted in this monosynaptic signal sparing preparation, and this monosynaptic response was inhibited by DAMGO in all 5 tested neurons (Supplementary Fig. 6). Therefore, we conclude that MORs are functionally expressed on LPO terminals that monosynaptically contact LHb neurons, and when these glutamatergic inputs are activated in vivo, DAMGO application should inhibit them.

**MOR mRNA is enriched in LHb-projecting LPO neurons**. Multiple basal forebrain structures express high levels of MOR including the VP, medial preoptic area (MPO), horizontal diagonal band (HDB), ventral bed nucleus of the stria terminalis (vBNST), and other regions of the extended amygdala complex (EAC)[40]. Some of these not only express MOR to a greater extent than the LPO, but they also project to the LHb. To further evaluate the specificity of MOR expression in LHb-projecting neurons in the LPO compared to nearby brain regions, we performed in situ hybridization for MOR mRNA (OPRM1) in brain slices from Sprague Dawley rats where the retrograde tracer Fluoro-Gold had been iontophoresed into the LHb (Fig. 4a–c). With this independent approach, the LHb-projecting LPO neurons showed the strongest OPRM1 expression and contained the greatest number of retrogradely labeled FG(+) neurons co-labeled for OPRM1 (112 ± 9 cells), corresponding to 57.1% ± 4.3% of all MOR(+)/FG(+) neurons in the basal forebrain (336/602 total cells). Inputs were also observed from the HDB (23.6% ± 5.9% of MOR(+) cells; 153/602 total cells), VP (9.5% ± 3.1% of MOR(+) cells; 52/602 total cells), MPO (3.0% ± 1.1% of MOR(+) cells; 19/602 total cells), vBNST (3.7% ± 1.2% of MOR(+) cells; 23/602 total cells) and EAC (3.0% ± 2.1% MOR(+) cells; 19/602 total cells; Fig. 4d–f). Overall, these

neurons in female rats was equivalent to that observed in males (Fig. 3f). Because of the prevalence of local glutamatergic connections in the LHb[18,35] (Supplementary Fig. 5) and postsynaptic MOR inhibition of a subset of LHb neurons[18], we sought to rule out a polysynaptic connection. First, a polysynaptic contribution seems unlikely for the glutamatergic inputs reported here because the delay from light pulse onset to EPSC onset was consistently <3 ms (Supplementary Fig. 4b). Second, to directly test isolated

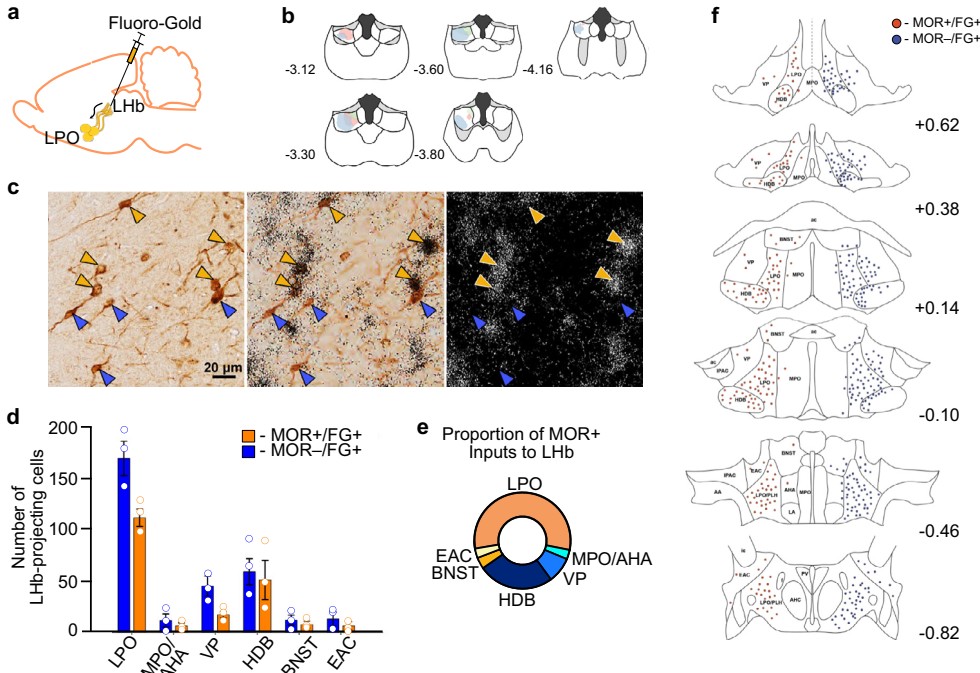

**Fig. 4 LPO inputs to the LHb highly express MOR mRNA. a** Schematic of unilateral iontophoretic deposition of the retrograde tracer FG into the LHb. **b** FG injection sites in LHb ($n = 3$ rats). **c** (Left and middle) Representative brightfield images of FG-positive/MOR-positive somas (orange arrowhead) and FG-positive/MOR-negative somas (blue arrowhead) in the LPO. Scale bar = 20 μm. (Right) Darkfield image of in situ hybridization localization for MOR mRNA (OPRM1). **d** Number of MOR-positive/FG-positive (orange) and MOR-negative/FG-positive (blue) cells in various basal forebrain regions (data are indicated as bar graphs with mean ± SEM, and circles as individual animals). **e** Donut plot depicting proportions of total MOR-positive inputs to the LHb from various basal forebrain regions. **f** Locations of FG-positive/MOR-positive (orange) and FG-positive/MOR-negative (blue) somata throughout basal forebrain. Co-labeled cells are separated from the singly labeled population for display purposes.

anatomical data are highly consistent with our electrophysiology results, supporting the conclusion that the LPO projection to the LHb is strongly regulated by MORs.

**LHb MOR activation blocks the aversiveness of LPO-LHb stimulation**. Stimulating LHb-projecting LPO glutamate neurons is aversive in mice[30]. Here we tested whether stimulating the LPO projection to the LHb in rats is aversive with a place conditioning assay. We used an intersectional viral approach injecting the retrograde CAV-Cre into the LHb and AAV2-EF1α-DIO-hChR2(H134R)-mCherry into the LPO bilaterally (Fig. 5a, b and Supplementary Fig. 8a). Bilateral optic fibers were implanted above the LPO for laser stimulation of these cell bodies. During conditioning sessions, rats received blue laser stimulation (473 nm, 20 Hz, 5 ms, 10–12 mW) in one environment and no stimulation in the other (Fig. 5c, top). Rats developed a conditioned place aversion to the laser-paired chamber (Fig. 5c, bottom).

Because ~75% of LHb-projecting LPO neurons in the rat are glutamatergic[30], and because we found that MOR activation inhibits the glutamatergic LPO terminals in the LHb, we hypothesized that the aversiveness produced by stimulating the LHb-projecting LPO neurons should be reduced by MOR agonist injection into the LHb. We tested this by combining the optogenetic intersectional virus manipulation above with behavioral pharmacology through bilateral cannulae aimed at the LHb (Fig. 5a, d and Supplementary Fig. 8a). During place conditioning sessions rats received either DAMGO or saline into the LHb while also receiving blue light activation of the LHb-projecting LPO neurons in both pairing environments (Fig. 5d, top). Rats with ChR2 expression developed a CPP for the chamber associated with DAMGO, while control mCherry-expressing animals did

not (Fig. 5d, bottom). Therefore, MOR activation in the LHb blocked the aversiveness of stimulating the LPO input to the LHb without producing positive reinforcement in the absence of input stimulation.

**Optogenetic inhibition of LHb-projecting LPO neurons in animals with chronic pain produces place preference**. To evaluate the direct contribution of LHb-projecting LPO neural activity to the aversiveness of ongoing pain, we selectively inhibited this circuit connection with the light activated chloride channel iC++ in rats with SNI and a similar intersectional virus approach (Fig. 6a, b and Supplementary Fig. 8b). Using ex vivo cell attached recordings in LPO neurons expressing iC++ we confirmed that light delivery inhibits action potential firing in these neurons (Fig. 6b). In the place conditioning assay, injured rats developed a preference for the laser-paired chamber but sham rats did not (Fig. 6c). Therefore, decreasing activity specifically in LHb-projecting LPO neurons when an animal is in pain is sufficient to relieve an aversive state and support a CPP.

**Noxious stimulation activates glutamatergic LHb-projecting LPO neurons**. Finally, we tested whether glutamatergic LHb-projecting LPO neurons are activated by noxious stimulation and whether ongoing pain alters this response. We expressed the calcium indicator GCaMP6m in LHb-projecting LPO neurons using a Cre-dependent, retrograde viral construct HSV-hEF1α-LS1L-GCaMP6m injected to the LHb of VGluT2::Cre mice (Fig. 6d). We implanted optic fibers above the LPO in control and SNI mice and performed fiber photometry prior to and during noxious heat stimulation. On average, GCaMP6m signal increased as paw withdrawal commenced, and this response was greatly potentiated in mice with SNI (Fig. 6e), who also displayed

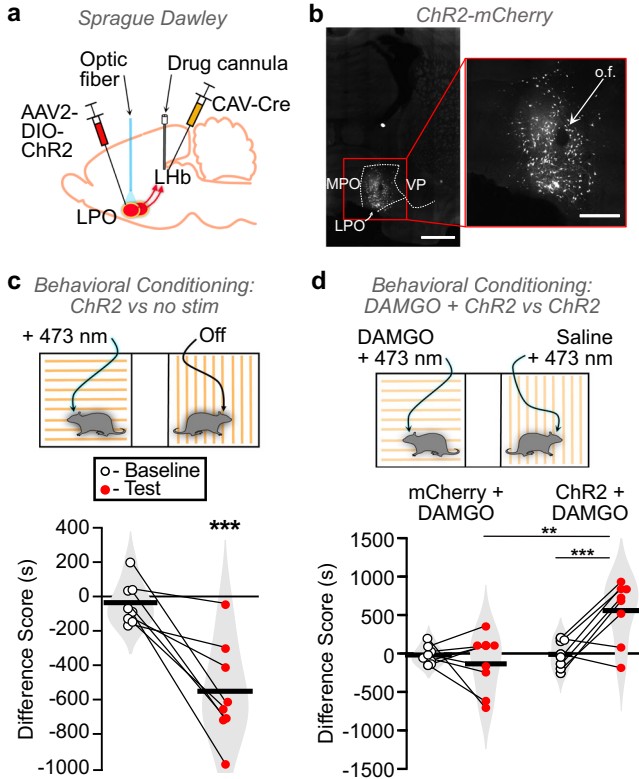

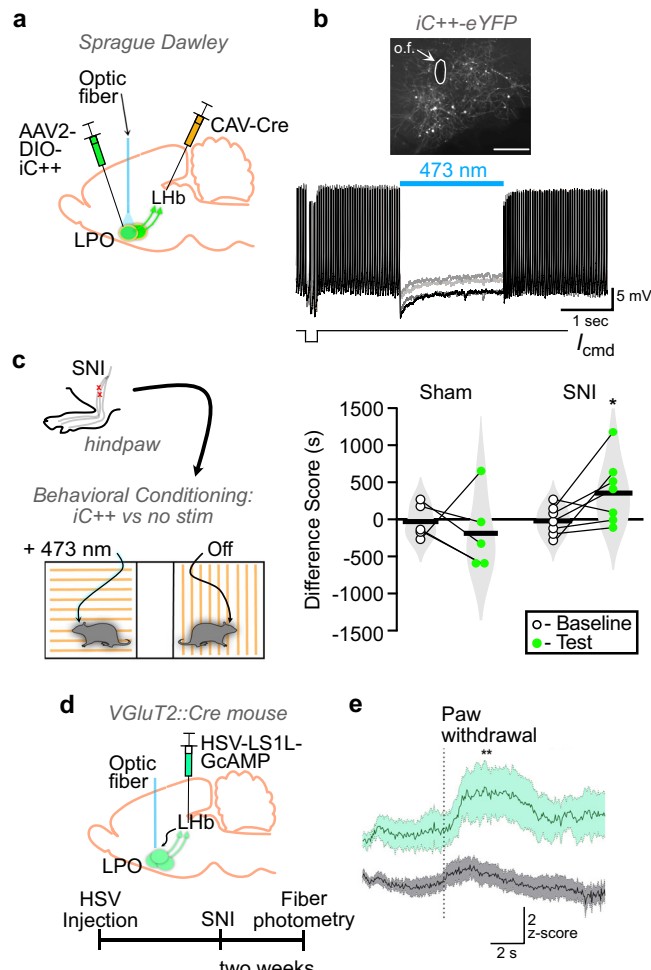

**Fig. 5 LHb MOR activation relieves the aversiveness of LPO-LHb stimulation. a** Schematic diagram of dual virus injections to express ChR2-mCherry or mCherry selectively in LPO neurons that project to the LHb, and optic fiber implantation targeting the LPO cell bodies in male rats. **b** Representative images of Cre-dependent ChR2-mCherry fluorescence in LPO cell bodies. (Left) Scale bar = 250 μm. (Right) Lesion from optic fiber ("o.f.") implant amidst ChR2-mCherry-expressing cell bodies. Scale bar = 50 μm. **c** Bilateral activation of the LPO-LHb projection (n = 8) produced a robust conditioned place aversion to the stimulation paired side of the place conditioning apparatus (paired *t*-test, df = 7, t = −4.64, p = 0.002). **d**, Rats with active ChR2 (n = 8) or mCherry controls (n = 8) received bilateral blue light stimulation in both conditioning chambers, concomitant with intra-LHb DAMGO in one conditioning chamber or intra-LHb saline in the other. Two-way mixed ANOVA, significant interaction between ChR2-mCherry/mCherry and baseline/test F(1,14) = 9.982, p = 0.007; post hoc effect group on test day adjusted p = 0.006; paired *t*-tests, mCherry adjusted p = 0.427; ChR2-mCherry adjusted p = 0.0027. Data was collected across a minimum of two cohorts of animals. Tests are two-tailed. *p < 0.05, **p < 0.01, ***p < 0.005.

shorter withdrawal latencies to heat (Supplementary Fig. 7). Thus, glutamatergic LPO neurons that project to the LHb are activated in response to noxious peripheral stimulation, and the magnitude of activation is greater during ongoing pain.

## Discussion

Here we identified a circuit that can be targeted by MOR agonists to relieve the aversiveness of ongoing pain but does not produce reward in pain-free rodents (Fig. 7). MOR agonist action in the LHb was sufficient to reverse injury-induced allodynia and to produce a CPP, decreasing both sensory and affective pain responses, respectively. Importantly, sham-injured animals did not develop a CPP, indicating that MOR activation in the LHb does not produce positive reinforcement in the absence of pain. Unexpectedly, rather than inputs from brain regions previously

**Fig. 6 Optogenetic inhibition relieves the aversiveness of pain-induced LPO-LHb activation. a** Schematic diagram of dual virus injections to express iC++-eYFP selectively in LPO neurons that project to the LHb, and optic fiber implantation targeting the LPO cell bodies in male rats that underwent either sham or spared nerve injury. **b** (Top) Representative image of cre-dependent iC++-eYFP fluorescence in LPO cell bodies. Scale bar = 50 μm. (Bottom) Representative cell-attached recording in current clamp mode of an LPO neurons expressing iC++-eYFP fluorescence, where blue light stimulation paused neuronal firing. **c** (Left) Sham (n = 5) or SNI (n = 7) rats were conditioned with bilateral blue light-induced inhibition in one chamber or no light in the other chamber. (Right) Difference scores before and after conditioning. Two-way mixed ANOVA, trending interaction between sham/spared nerve injury (SNI) and baseline/test F(1,10) = 3.443, p = 0.093; post hoc paired *t*-tests for sham adjusted p = 0.580; spared nerve injury (SNI) adjusted p = 0.048. **d** Schematic diagram of HSV-hEF1α-LS1L-GCaMP6m injection into the LHb of VGluT2::Cre mice, to induce expression of GCaMP6m in glutamatergic LPO neurons innervating the LHb, and optic fiber implant for fiber photometry imaging in the LHb-projecting LPO neurons. (Bottom) Experimental timeline. **e** Mice with SNI (green, n = 5) exhibited significantly larger changes in GCaMP6m fluorescence time-locked to paw withdrawal in response to thermal stimulation in the Hargreaves apparatus compared to sham controls (gray, n = 9): Two-way ANOVA, F(1,12) = 5.439, p = 0.038; Holm-Sidak post-hoc test, p = 0.0074. Data were collected across a minimum of two cohorts of animals. Tests are two-tailed. *p < 0.05, **p < 0.01.

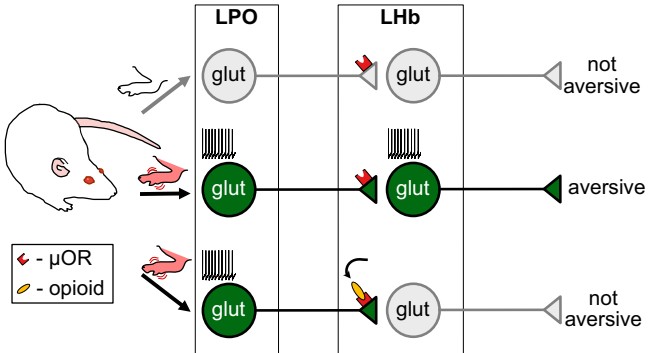

**Fig. 7 Summary: LHb-projecting LPO glutamatergic neurons that fire more during pain provide an aversive signal.** MOR activation in the LHb inhibits the glutamate release from the LPO, preventing propagation of the aversive signal further in the circuit.

established as mediating pain and opioid-induced pain relief, it is the glutamatergic inputs from the LPO that we demonstrate are preferentially controlled by LHb MOR activation. We found that the aversiveness produced by optogenetic activation of LHb-projecting LPO neurons is blocked by MOR agonist microinjection into the LHb. Importantly, optogenetic inhibition of this circuit in injured animals produced place preference, parallel to LHb MOR activation and indicating that inhibiting LHb-projecting LPO neurons is sufficient to relieve the ongoing aversive state. Together, these experiments show that MOR activation in the LHb can generate negative reinforcement via pain relief, but not positive reinforcement in the absence of noxious stimuli.

**The LHb in pain and relief.** The LHb plays a role in the perception of noxious stimuli in injury and depression models[41,42], and most LHb neurons have increased firing rates during aversive behavioral states and in response to noxious stimuli[2,3]. Bilateral lesions of the LHb decrease hypersensitivity to touch generated by the chronic constriction ischemia model of neuropathic pain in rats[43], supporting the notion that signaling through the LHb contributes to injury-induced mechanical allodynia. In the aversive state of alcohol withdrawal, chemogenetic inhibition of the LHb relieves thermal hyperalgesia[41]. We found that LHb MOR activation reverses mechanical hypersensitivity in a model of neuropathic pain in males but not significantly in females. In place conditioning measures, in a design used to evaluate the impact of a manipulation on the affective component of pain, we observed that intra-LHb DAMGO induced a CPP in both males and females with ongoing pain, but not in shams of either sex. Female rats required a higher concentration of DAMGO to produce intra-LHb CPP, consistent with prior studies showing that across rodent strains females show less opioid-induced antinociception than males[44,45]. Together these observations support the model that LHb neural activity increases in aversive behavioral states and manipulations that decrease firing provide a relief signal.

We previously discovered two mechanisms by which MOR activation inhibits LHb neural activity: presynaptic inhibition of glutamate release or postsynaptic outward current generation in a subset of LHb neurons[18]. Here we found that these two MOR effects are intact in animals with ongoing pain. Because the behavioral effect of LHb MOR activation was specific to injured rats we hypothesized that in control conditions the MOR sensitive LHb circuit elements have low activity levels and this activity increases in ongoing pain. Since DAMGO induced outward currents in LHb neurons are modest and DAMGO-

induced inhibition of glutamate release onto a subset of LHb neurons was quite strong, we further hypothesized that there is increased glutamatergic drive to the LHb in the pain state, in an input that is MOR sensitive.

The vast majority of LHb neurons are glutamatergic, and there is anatomical evidence for dense bouton-like structures arising from local LHb neurons[46]. This local feed forward connectivity may distribute a specific excitatory LHb input across the various LHb projections that include the dorsal raphe, ventrolateral PAG, MDL, centromedian thalamus, LH, RMTg, and VTA[9,13,14,47]. Local connectivity has also been observed with functional assays, including that TTX application decreases the frequency of sEPSC and sEPSP events in a subset of LHb neurons[18,35] and that optogenetic activation of LHb neurons induces glutamatergic EPSCs (Supplementary Fig. 5). This feed forward circuit enables the distribution of an afferent excitatory signal across LHb neurons, and it raises the possibility that inhibition of such an input by a MOR agonist will decrease activity in many LHb projection neurons, even those that are not directly innervated by the excited pathway. Therefore, it is possible that MOR inhibition of one specific glutamatergic input, such as the LPO, can decrease the distribution of an excitatory drive onto many efferent LHb projections.

We characterized the functional glutamatergic connections from a variety of brain regions to the LHb in rats; our findings are largely consistent with prior reports utilizing similar techniques in mice. Still, we note that outcomes of these experiments are dependent on the types and number of neurons that express ChR2 following the virus injections and are limited by the geometry of an injection site. We detected the strongest glutamatergic inputs from the LH and EPN, while glutamatergic synaptic responses from the VTA were quite small. We found no functional inputs from the ACC, and in a systematic anatomical analysis we detected very few afferent fibers from the ACC present in the LHb. Of the detected glutamatergic inputs, we expected brain regions associated with pain perception and pain relief, such as the LH and VTA, to be more strongly modulated by MOR activation. However, we instead found MOR mRNA and function were clearly enriched in LPO inputs to the LHb.

Because LHb neurons generally fire more in response to noxious stimuli, and optogenetic stimulation of various glutamatergic inputs to the LHb is uniformly aversive[27,29,30,48,49], we hypothesized that a glutamatergic input to the LHb transmits the pain signal. Further, aversive stressors lead to an increase in the ratio of excitatory glutamatergic to inhibitory GABAergic synaptic input to LHb neurons[48–50], and restoration of this ratio is associated with relief of aversive states such as foot shock-induced learned helplessness[49] and cocaine withdrawal[48]. Therefore, a pharmacological manipulation that decreases the excitatory drive onto LHb neurons should also relieve aversive states. Because we found that DAMGO in the LHb generates CPP only in animals with ongoing pain, our data suggest a MOR-sensitive glutamatergic input to the LHb is active during pain and relatively inactive in the absence of pain.

**The LPO: a brain region contributing to pain perception.** The LPO projection to the LHb is composed of neurons releasing either glutamate or GABA; these neurons do not co-release glutamate and GABA[30], unlike other LHb inputs[51]. Optogenetic activation of LPO glutamate projections to the LHb is aversive in mice[30], and here we posited that this activation mimics an ongoing pain signal. We found in rats that stimulating the LPO inputs to the LHb even without isolating the glutamatergic component is also aversive. This is consistent with our electrophysiology observations that the glutamate component of this projection appears stronger than the

GABAergic component in our preparations. Isolated activation of the VGaT LHb-projecting LPO neurons generates a place preference[30], thus stimulating both populations of neurons in rats here may generate a weaker aversion than selective activity in the glutamatergic component of the connection. However, GCaMP6s recordings in transgenic mice show that activity increases in response to an aversive stimulus and its predictive cue in both VGaT and VGluT2 LPO neurons that project to the LHb[30], therefore stimulating both inputs may have some relation to in vivo activity patterns. We were able to relieve the aversiveness of activation of this connection with a MOR agonist in the LHb. Further, inhibiting the LPO projection to the LHb specifically in injured animals was sufficient to produce a CPP. While a causal role for the LPO in pain perception is unexplored to date, anterograde[52] and retrograde[53] tracing has shown a direct input to the LPO from the spinal cord. Also, injections of the pro-inflammatory cytokine IL-1ß into the LPO induces hyperalgesia, indicating LPO participation in a nociception circuit[54]. Painful stimuli such as subcutaneous formalin injections, mild electric shock, and tail pinch also increase firing in some LPO neurons[55,56]. Here we show that glutamatergic LPO projections to the LHb increase their activity in response to noxious stimulation and that the magnitude of this response is greater in animals with ongoing pain. This combined with our optogenetic manipulations indicate that a major output for LPO pain signals is to the LHb.

**The LPO to LHb circuit: a unique target for analgesia**. Opioids remain the best available clinical analgesics, yet systemically administered opioids can result in serious adverse consequences including opioid use disorder and respiratory depression[57]. Opioid-induced positive reinforcement and euphoria, combined with the development of dependence, underlie opioid abuse liability. To date, there is little clinical evidence that the analgesic and euphoric effects of opioids can be decoupled in humans, yet this possibility is a potential pathway to improve therapies for pain. Here we have identified a key circuit whose modulation relieves pain but does not generate reward in the absence of pain. This dissociation between relief and reward provides a neural target for pain treatment without promoting substance use disorder. Is it possible to activate MORs in the LHb but not in a reward circuit? There is some preclinical evidence that this dissociation may be achieved by developing MOR ligands with the appropriate opioid pharmacology. For example, a recently developed cyclized, stabilized MOR selective agonist based on endomorphin I produces pain relief but not reward[58]; a similar pattern has been observed for analogs of mitragynine[59]. An alternative approach would be to target a different receptor with high expression levels in this circuit in a way that would decrease LHb neural activity. Relevant to this approach, mRNA expression for a selection of orphan G-protein coupled receptors (GPCRs) is enriched in the habenula[39,60,61]. One such receptor is GPR151, and LHb neurons containing GPR151 receive input from the LPO[62]. Thus, directed strategies like these, including a focus on the LPO-LHb circuit, present a range of possible anatomic and molecular targets for pain therapy.

## Methods

**Animals**. All experiments were performed in accordance with the guidelines of the National Institutes of Health Guide for the Care and Use of Laboratory Animals and the Institutional Animal Care and Use Committees (IACUC) at the University of California San Francisco, the National Institute on Drug Abuse (NIDA), and Rutgers University.

Male and female Sprague Dawley rats were obtained from Charles River Laboratories. Rats were allowed access to food and water ad libitum and maintained on a 12 h:12 h light/dark cycle. Rats used in behavioral and in situ hybridization studies were housed under reverse light/dark cycle conditions. Rats were group housed until they underwent surgery, after which they were singly housed.

Male and female VGluT2::Cre mice were bred from mice obtained from the Jackson Laboratory (Jax # 016963). Mice were allowed access to food and water ad libitum and maintained on a 12 h:12 h light/dark cycle with lights on at 7 AM. Mice were always housed in groups of 2–5.

**Viral constructs**. AAV2-hSyn-hChR2(H134R)-mCherry (titer: 2.9e + 12), AAV2-hSyn-mCherry (titer: 4.7e + 12), and AAV2-EF1α-DIO-ChR2-mCherry (titer: 5.1e + 12), and AAV2-Ef1α-DIO-iC + +-eYFP (titer: 5.9e + 12) were purchased from the University of North Carolina Vector Core with available stock constructs from the laboratory of K. Deisseroth at Stanford University. CAV-Cre (titer: 2.5e + 12) was purchased from Montpelier University, France. HSV-hEF1α-GCaMP6m (titer: 5e + 9) was purchased from the Gene Delivery Technology Core at Massachusetts General Hospital.

**Stereotaxic injections**. Rats weighing 275–300 g were anesthetized with 3–5% isoflurane (Henry Schein) via inhalation and secured in a stereotaxic frame. Bilateral craniotomies were created with a dental drill above the injection site. For electrophysiology experiments, injections of AAV2-hSyn-hChR2(H134R)-mCherry were made bilaterally into the LPO (−0.3 mm anteroposterior (AP), ± 1.4 mm mediolateral (ML), −8.4 mm dorsoventral (DV)), VP (−0.24 mm AP, ± 2.6 mm ML, −7.8 mm DV), EPN (−2.4 mm AP, ±3.0 mm ML, −7.0 mm DV), LH (−2.6 mm AP, ±1.7 mm ML, −8.2 mm DV), VTA (−5.8 mm AP, ±0.5 mm ML, −8.5 mm DV), anterior ACC (+2.2 mm AP, ±0.6 mm ML, −2.6 mm DV), or posterior ACC (+1.7 mm AP, ±0.6 mm ML, −2.6 mm DV) using a Nanoject II (Drummond Scientific, Broomall, PA). A volume of ~500–830 nL was injected per hemisphere over a period of 4.5 min. The glass injector tip was left in place for at least 2 additional minutes before slow withdrawal to prevent backflow and infection of tissue dorsal to the injection target.

For optogenetic behavior experiments, AAV2-EF1α-DIO-hChR2(H134R)-mCherry, AAV2-EF1α-DIO-mCherry, or AAV2-EF1α-DIO-iC++-eYFP were injected into the LPO as above and CAV-Cre was injected bilaterally into the LHb (−3.6 mm AP, ±0.65 mm ML, −5.4 mm DV) with a microinjector connected via polypropylene tubing to Hamilton syringes controlled by a dual syringe pump (KD pump) guided by a bilateral 33 G stainless steel guide cannula (Plastics One).

Rats were treated with subcutaneous Carprofen (5 mg/kg, Zoetis) and topical 2% Lidocaine (Phoenix Pharmaceutical, Inc.) during the surgery for pain control. After surgery, animals had access to liquid Tylenol (~1:40) in their drinking water for 3–5 days or were administered Meloxicam (s.c. 2 mg/kg, Pivetal) once per day for two days.

**Cannulation and optic fiber implantation**. Two to four weeks after virus injection, rats slated for behavioral testing underwent a second cranial surgery to implant custom-made 200 μm optic fibers at a 5° angle of rotation in the coronal plane into the bilateral LPO (−0.6 mm AP, ±3.05 mm ML, −7.5 mm DV). For microinjections into the LHb, bilateral guide cannulae were implanted 1 mm above the LHb (−3.7 mm AP, ±0.65 mm ML, −4.4 mm DV). A dummy stylet was inserted to maintain patency of the cannulae. Optic fibers and cannulae were anchored with flat point screws and dental cement. For i.c.v. microinjections, unilateral cannulae were implanted into the right lateral ventricle (−1.0 mm AP, +1.5 mm ML, −3.5 mm DV).

Analgesia during surgery and recovery was administered as described above. Animals were allowed to recover for 1–2 weeks prior to behavioral testing. All virus injections, optic fiber placements, and cannulae placements were histologically verified postmortem based on the standard rat brain atlas[63].

**Spared nerve injury**. Spared nerve injury (SNI) of the sciatic nerve branch was performed to model chronic neuropathic pain[19]. Under isoflurane anesthesia, a 2-cm skin incision was made over the left hindlimb. The biceps femoris muscle was blunt dissected to expose the branches of the sciatic nerve. The common peroneal and tibial nerves were ligated with 5.0 silk surgical suture and transected distally, with sparing of the sural nerve branch. For sham procedures, a skin incision was made and biceps femoris muscle was exposed without dissection. The overlying skin was closed with a monocryl suture. Animals were allowed seven days to recover from surgery prior to behavioral testing. Behavioral indications of ongoing pain such as paw guarding were observed in all animals used in the SNI groups.

**Inflammatory pain model**. Peripheral inflammation was induced using Complete Freund's Adjuvant (CFA; Sigma Life Science). Under isofluorane anesthesia, a 1:1 emulsion of CFA and sterile saline (150 μL) was injected into the footpad of the rat's left hindpaw with a 27G needle. Sham-injured controls were injected with sterile saline (150 μL).

**Microinjections**. Rats were lightly restrained in a cloth wrap for intracranial microinjections. A bilateral 33 G microinjector (PlasticsOne) that extended 1 mm ventrally beyond the guide cannula was inserted to target drug delivery into the LHb or i.c.v. Hamilton syringes were driven by a dual syringe pump to infuse either

vehicle (phosphate-buffered saline, PBS) or DAMGO (10 μM, 300 nL/hemisphere administered over 2 min). A separate cohort of female rats was microinjected with 100 μM DAMGO.

**von Frey sensory assessment**. Standard von Frey sensory assessments were performed as described[64]. Briefly, rats were habituated to sensory testing chambers (Plexiglass boxes with mesh-like flooring) for at least 2 days prior to testing. Behavioral assessments did not begin until exploratory behavior subsided. Testing was completed with eight Touch Test® fibers (North Coast Medical & Rehabilitation Products, Gilroy, CA, USA) ranging from 0.4 to 15 g. Fibers were pressed perpendicularly to the mid-plantar left hindpaw with sufficient force to cause bending in the fiber and held for 3–4 s. A positive response was noted if the paw was sharply withdrawn. Ambulation was not considered a positive response. When responding was ambiguous, testing was repeated. 50% withdrawal thresholds were calculated using fiber application in ascending stiffness order or using the up down method.

In behavioral pharmacology experiments rats underwent sensory testing 5 min after counterbalanced saline or DAMGO microinjections on the same day, with at least 4 h between infusions. Experimenters were blinded to the solution composition during administration and testing.

**Hargreaves sensory assessment**. Hargreaves tests were completed in sensory testing chambers with glass flooring. Assessment commenced after rats acclimated to the chamber. Testing was completed using a plantar test analgesia meter (Series 8 Model 390, IITC Life Science, Woodhills, CA, USA). Radiant light was directed toward the mid-plantar left hindpaw until a sharp paw withdrawal response was observed with a 30-s maximum cutoff. Ambulation was not considered a positive response. At baseline testing, measurements were repeated at different intensity levels until the average withdrawal latency of eight trials was $15 \pm 2$ s. Subsequent measurements were performed using this individualized intensity level.

**Place conditioning**. Conditioned place preference (CPP) pairings occurred twice daily for 4 consecutive days following the post-surgery sensory tests with DAMGO and saline. The conditioning boxes (Med. Associates, Georgia, VT, USA) have three divisions: two conditioning chambers (25 cm × 21 cm × 21 cm) with distinct visual (horizontal vs. vertical stripes) and textural (thick vs. thin mesh flooring) cues, separated by a third, smaller gray neutral chamber (12 cm × 21 cm × 21 cm). Before conditioning commenced, animals were allowed up to three opportunities to show neutrality across the chambers during 30-min baseline sessions. Rats that displayed a consistent baseline preference (>65% of time spent in one chamber) were excluded from the study. Rats were pseudorandomly assigned to receive DAMGO in one of the conditioning chambers, and assignments were counter-balanced for each cohort.

During conditioning sessions, microinjections were performed as described above through the intra-LHb cannulae just before the rat was confined to the designated chamber for 30 min. Morning and afternoon pairing sessions were at least 4 h apart and the order of pairings was alternated on each conditioning day. On test day, rats were allowed to freely explore the chambers for 30 min and time spent (s) in each partition was recorded. Difference score was defined as (Time spent in DAMGO-paired chamber)−(Time spent in saline-paired chamber).

For ChR2 activation studies, rats were acclimated to handling and attachment of fiber cables to fiber implants in a neutral environment. On procedure days, fiber implants were connected to optic fiber cables attached to a 1 × 2 fiber optic rotary joint (Doric Lenses, Quebec, Canada). A laser light source (MBL 473, OEM Laser Systems, East Lansing, MI) was adjusted to deliver 80–120 mW/mm light intensity at the end of the fiber implant[2]. Light stimulation (5 ms pulses at 20 Hz) commenced upon placing rats into the designated chamber of a custom built three chamber box on conditioning days. During baseline and testing sessions, the time spent in each chamber was recorded using a webcam and analyzed using Viewer software (Biobserve, Bonn, Germany).

**Brain removal and immunohistochemistry**. Rats were deeply anesthetized with an intraperitoneal injection of Euthasol (0.1 mg/kg, Virbac Animal Health, Fort Worth, TX) after 0.3 μL of Chicago Sky Blue (in 2% PBS) was injected through the cannulae to mark injection locations. After becoming unresponsive to noxious stimuli, the rats were transcardially perfused with 400 mL of saline, followed by 400 mL of 4% paraformaldehyde (PFA) in 0.1 M phosphate buffer. The brains were extracted and immersion-fixed in PFA for 2 h at room temperature (RT), washed two times with PBS to remove excess PFA, and stored in 1X PBS at 4 °C until they were sectioned (50 μm) using a vibratome (Leica VT 1000S). Sections were mounted on slides using VECTASHIELD® mounting medium (Vector Laboratories, Burlingame, CA, USA). Images were taken under a Zeiss Stemi 2000-C (Pleasanton, CA) using an Amscope MD800E running AmScope x64 3.0 Imaging Software.

To label biocytin filled cells after slice electrophysiology recordings, slices were washed three times for 5 min each with PBS (Gibco, Waltham, MA), then blocked with a solution containing: bovine serum albumin (0.2%), normal goat serum (5%) and Tween20 (0.3%; Sigma-Aldrich, St. Louis, MO) for 2 h at RT. Slices were incubated in DTAF-streptavidin (1:200; Jackson Immuno Research) diluted in

PBS + 0.3% Tween20 for 48 h at 4 °C. After five, 10-min rinses, brain slices were mounted onto glass slides as above and imaged using a Zeiss Axioskop upright microscope (2.5X, NA = 0.075 or Plan Apochromat 20X, NA = 0.75).

**Electrophysiology**. Rats were deeply anesthetized with isoflurane, decapitated, and brains were quickly removed into ice-cold artificial cerebrospinal fluid (aCSF) consisting of (in mM): 119 NaCl, 2.5 KCl, 1.0 NaH$_2$PO$_4$, 26.2 NaHCO$_3$, 11 glucose, 1.3 MgSO$_4$, 2.5 CaCl$_2$, saturated with 95% O$_2$-5% CO$_2$, with a measured osmolarity 310–320 mOsm/L. Two hundred μm coronal sections through the LHb were cut with a Leica VT 1000 S vibratome. Tissue containing the virus injection sites was drop fixed in 10% formalin. Slices were incubated in oxygenated aCSF at 33 °C and allowed to recover for at least one hour. A single slice was placed in the recording chamber and continuously superfused at a rate of 2 mL/min with oxygenated aCSF. Neurons were visualized with an upright microscope (Zeiss AxioExaminer.D1) equipped with infrared-differential interference contrast, Dodt optics, and fluorescent illumination. Whole-cell recordings were made at 34 °C using borosilicate glass microelectrodes (3–5 MΩ) filled with K-gluconate internal solution containing (in mM): 123 K-gluconate, 10 HEPES, 8 NaCl, 0.2 EGTA, 2 MgATP, 0.3 Na$_3$GTP, and 0.1% biocytin (pH 7.2 adjusted with KOH; 275 mOsm/L). Liquid junction potentials were not corrected during recordings. Input and series resistance were monitored throughout voltage clamp experiments with a −4 mV step every 30 s. Series resistance was required to be 5–30 MΩ and cells with series resistance changes >25% were excluded.

Signals were recorded using a patch clamp amplifier (Axopatch 1D, Molecular Devices, San Jose, CA or IPA, Sutter Instruments, Novato, CA). Signals were filtered at 5 kHz and collected at 20 kHz using IGOR Pro (Wavemetrics) or collected at 10 kHz using SutterPatch software (Sutter Instruments). Light-evoked EPSCs and IPSCs were evoked by two blue light pulses (473 nm, 1–10 ms) administered 50 ms apart, once every 30 s. LHb recordings were generally made in LHb subregions enriched in ChR2-expressing fibers. Recordings were made in voltage-clamp mode, with membrane potential clamped at $V_m = -60$ mV and −40 mV, for EPSCs and IPSCs respectively. Light was delivered by an LED coupled to an optic fiber aimed at the recorded cell (7–10 mW). To calculate connectivity rates, only the first neuron patched per slice was included in order to avoid over sampling from slices or animals with lower infection rates. All measurements of DAMGO effects on EPSCs were completed in the presence of gabazine. After recordings, slices were drop fixed in 4% PFA for at least 2 h at 4 °C and processed for biocytin labeling.

*Data analysis for electrophysiology.* Light pulses were considered to reveal synaptic connections when three conditions were met: (1) the average of 8 traces showed a deviation from baseline $I_{holding}$ such that the mean trace exceeded 4 SD of 10 ms baseline period within the 10 ms window after initiation of light pulse, (2) the putative response was observed in at least three independent trials, and (3) the delays from the light stimulation onset of putative responses were time locked (<1 ms jitter) across trials. Latency was calculated as time from start of light pulse to when the rate of rise exceeded −40,000 V/s. In some cases DNQX (10 μM) or gabazine (10 μM) was bath applied to confirm inward and outward currents as AMPA receptor or GABA$_A$ receptor-mediated, respectively. The magnitude of DAMGO induced changes were quantified by comparing the mean values over the last 4 min of DAMGO application to the 4 minutes of baseline just prior to starting the DAMGO application. Where individual neurons were evaluated for significant responses, DAMGO sensitive neurons were identified by comparing 30 s binned samples during these two time windows with an unpaired *t*-test.

**Combined retrograde tracing and in situ hybridization**

*Tracer injections.* Male Sprague Dawley rats (300–500 g) were anesthetized with 2–5% isoflurane. 1% Fluoro-Gold (FG; FluoroChrome LLC) solution in a 0.1 M cacodylate buffer (pH 7.5) was delivered unilaterally into the LHb (−3.4 mm AP, ±0.9 mm ML, and −5.4 mm DV) iontophoretically through a stereotaxically positioned glass micropipette (18–25 μm inner diameter) by applying 1 μA, 7 s pulses at 14 s intervals for 20 min. The micropipette was then left in place for an additional 10 min to prevent backflow. Following surgery, rats were singly housed and perfused 3 weeks later.

*Tissue preparation.* Rats were anesthetized with chloral hydrate (0.5 ml/kg) and perfused transcardially with 4% (w/v) PFA in 0.1 M phosphate buffer treated with diethylpyrocarbonate (DEPC), pH 7.3. Brains were post-fixed in 4% PFA for 2 h before being transferred to an 18% sucrose solution (w/v in 0.1 M PBS) and stored overnight at 4 °C. Coronal sections of the LHb (30 μm) and LPO (16 μm) were prepared.

**Phenotyping of retrogradely labeled cells by immunocytochemistry and in situ hybridization**. Sections in the LPO were incubated for 2 h at 30 °C with rabbit anti-FG antibody (1:500; AB153; Millipore) supplemented with RNAsin. Sections were then incubated in biotinylated goat anti-rabbit antibody (1:200; BA1000; Vector Laboratories) for 1 h at 30 °C. Sections were then rinsed and treated with 0.2 N HCl, rinsed, and then acetylated in 0.25% acetic anhydride in

0.1 M triethanolamine. Subsequently, sections were rinsed and post-fixed with 4% PFA, rinsed, and then incubated in a hybridization buffer for 2 h at 55 °C.

Hybridization was then performed for radioactive detection of MOR mRNA by hybridizing sections for 16 h at 55 °C with [35 S]- and [33 P]-labeled (107 c.p.m./mL) single-stranded antisense probes. Following hybridization, sections were treated with 4 μg/mL of RNAse A at 37 °C for 1 h, washed with 1X saline-sodium citrate and 50% formamide for 1 h at 55 °C, and then with 0.1X saline-sodium citrate at 68 °C for 1 h. To visualize FG(+) cells, sections were rinsed with PBS and incubated for 1 h at RT in avidin-biotinylated horseradish peroxidase (1:100, ABC kit; Vector Laboratories). Sections were then rinsed, and the peroxidase reaction was developed with 0.05% 3,3'-diaminobenzidine tetrahydrochloride (DAB) and 0.003% $H_2O_2$. Sections were then photographed under bright field illumination and mounted on coated slides. Finally, slides were dipped in Ilford K.5 nuclear tract emulsion (Polysciences; 1:1 dilution in double-distilled water) and exposed in the dark at 4 °C for 3–4 weeks before development and photographs of silver-grain epiluminescence.

**Data analysis of in situ hybridization studies**. Methods for analysis of in situ hybridization material have been described previously[51]. Briefly, pictures were adjusted to match contrast and brightness by using Adobe Photoshop (Adobe Systems). Cell counting was completed independently by three scorers blind to the hypothesis of the study. Radioactive in situ material was analyzed using epilumi-nescence to increase the contrast of silver grains as described previously[65]. FG(+) cells (detected by fluorescence and brown DAB-label) were evaluated for the presence of MOR mRNA: a cell was considered to express MOR mRNA when its soma contained concentric aggregates of silver grains that exceeded background levels.

**Fiber photometry**

*Surgery*. Male and female VGluT2::Cre mice (20–30 *g*; 6–12 weeks) were anes-thetized with 1–5% isoflurane and secured to a stereotaxic frame. Using a Micro4 controller and UltraMicroPump, 0.2 μL of a retrograde, Cre-dependent HSV encoding GCaMP6m (HSV-hEF1α-LS1L-GCaMP6m) was injected into the LHb (−1.5 mm AP, +0.45 mm ML, −3.0 mm DV). Syringes were left in place for 7–10 min following injections to minimize diffusion. For fiber photometry calcium imaging experiments, a 400 μm core optic fiber (Doric Lenses) embedded in a 2.5-mm ferrule was implanted over the LPO (+0.5 mm AP, +0.8 mm ML, −5.05 mm DV) and secured to the skull using #000 screws (Fasteners and Metal products Corp; #000-120 ×1/16) and dental cement. Following surgery, mice recovered on a warm heating pad before being transferred back to the vivarium home cage. Three weeks after the virus and fiber surgery, mice were given either SNI or sham control surgery as described above.

*Recording*. Signals from GCaMP6 were recorded across 10 trials of stimulation in the Hargreaves test using a Plantar Test Instrument. The onset and offset time for each trial was digitized and sent to an RZ5D (Tucker Davis Technologies). For the acquisition of LPO→LHB activity, GCaMP6 was excited at two wavelengths (490 nm, calcium-dependent signal and 405 nm isosbestic control) by amplitude modulated signals from two light-emitting diodes reflected off dichroic mirrors and coupled into a 400 μm 0.48NA optic fiber. Signals emitted from GCaMP6m and its isosbestic control channel then returned through the same optic fiber and were acquired using photoreceiver (Doric Lenses), digitized at 1 kHz, and then recorded by a real-time signal processor (RZ5D; Tucker Davis Technologies) running the Synapse software suite. Analysis of the resulting signal was performed as described in Bruno et al. using custom-written MATLAB scripts available in a general release form at https://github.com/djamesbarker/FiberPhotometry [66–68]. Briefly, changes in fluorescence across the experimental session (ΔF/F) were cal-culated by smoothing signals from the isosbestic control channel, scaling the iso-sbestic control signal by regressing it on the smoothed GCaMP signal, and then generating a predicted 405 nm signal using the linear model generated during the regression. Calcium independent signals on the predicted 405 nm channel were then subtracted from the raw GCaMP signal to remove movement, photo-bleaching, and fiber bending artifacts. Signals from the GCaMP channel were then divided by the control signal to generate the ΔF/F. Peri-event histograms were created by averaging changes in fluorescence (ΔF/F) across repeated trials during windows encompassing behavioral events of interest. The area under the curve (AUC) was calculated for a pre-stimulation 5 s baseline commencing −10 s before paw withdrawal and for the 5 s period initiated with paw withdrawal.

**Anterograde tracing from the ACC**. In male rats, unilateral injections of AAV2-hSyn-hChR2(H134R)-mCherry (759 nL) were made throughout anteroposterior range of the ACC (+2.6 to −0.4 mm AP, −0.4 to −0.5 mm ML, −2.6 to −2.8 mm DV). Rats were perfused and brains fixed five weeks later, as described above. Coronal sections (50 μm) containing the ACC and the LHb were collected. After verification of the injection site, every sixth slice containing the LHb was rinsed twice with PBS. Tissue was pre-permeabilized in 1:1 EtOH:PBS for 30 min at 4 °C, rinsed briefly in PBS before blocked in 3% $H_2O_2$ for 10 min. Following PBS washes (3 × 5 min), tissue was blocked in solution containing normal goat serum (NGS, 10%) for 1 h at RT. Slices were incubated in rabbit anti-mCherry antibody (1:5000

in PBS + 0.3% Triton X100+NGS 10%; Abcam) overnight at 4 °C. After PBS washes (4 × 10 min), slices were incubated in biotinylated goat anti-rabbit sec-ondary antibody (1:200 in PBS; Vector Laboratories) for 2 h at 4 °C. Following PBS wash (4 × 10 min), slices were incubated in VECTASTAIN® ABC Reagent (VEC-TASTAIN® ABC Kit, Vector Laboratories) for 30 min followed by peroxidase substrate (DAB Substrate Kit, Vector Laboratories) for 10 min. Once dark brown DAB precipitate formed, slices were rinsed in PBS (5 × 5 min), mounted on glass slides, and cover slipped with DEPEX (Electron Microscopy Sciences). DAB-stained fibers were visualized under brightfield illumination and quantified in Stereo Investigator software (MBF Bioscience) using the virtual isotropic space balls probe[69].

**General experimental design**. For behavioral experiments, subject numbers were determined by pilot studies and power analyses (power = 0.80, significance level = 0.05, effect size = 15–30%). All behavioral experiments were performed blinded to experimental condition. For immunohistological experiments, three animals with injections targeted at the anterior, middle, and posterior ACC were used to obtain a comprehensive estimation of fibers projecting to the LHb. Electro-physiology experiments were conducted blind to injection site. No a priori power analysis was conducted for either immunohistological or electro-physiological experiments.

**Quantification, statistical analysis, and reproducibility**. Data are expressed as mean ± SEM or mean with 25th and 75th percentiles as indicated in figure legends and text. Significance was set at $p < 0.05$. Datasets were evaluated to determine whether parametric or non-parametric statistical approaches were most appro-priate as indicated in Supplemental Table 1. All tests were two tailed, and statistical analyses were performed in GraphPad Prism or R. The bandwidth for violin plots was determined by Silverman's rule of thumb in Plotly for Python. Sample sizes are reported in figure panels, legends, and Supplementary Table 1. Outliers, including extreme outliers, are reported in Supplementary Table 1 and were not removed from datasets.

**Reporting summary**. Further information on research design is available in the Nature Research Reporting Summary linked to this article.

## Data availability
The data generated in this study have been deposited in the Open Science Framework database [https://osf.io/mwyb3/?view_only=28fcc13c3cac41a7a55e27ffccd62bda]. The processed data are available at the Open Science Framework database. The data generated in this study are also provided in the Supplementary Information/Source Data file. Source data are provided with this paper.

## Code availability
Custom Matlab scripts for fiber photometry analysis can be found here https://github.com/djamesbarker/FiberPhotometry

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

## Acknowledgements

The authors would like to thank Ryan Carothers, Gabrielle Mintz, Lucy He, Venkateswaran Ganesh, and Benjamin Snyder for their technical assistance with histology, stereotaxic surgeries, and behavioral studies. We also thank Allan Basbaum for feedback on the manuscript. This work was supported by National Institutes of Health grants R01DA042025 (to E.B.M.), K08 NS097632 (to M.W.W.), the Intramural Research Program (IRP) of the National Institute on Drug Abuse (IRP/NIDA/NIH), and a NIDA K99/R00 pathway to independence award (DA043572) to D.J.B.

## Author contributions

E.B.M. conceived and supervised research. E.B.M., M.W.W., and D.J.B. designed the experiments. M.W.W., K.A.M., J.R.D., K.A.M., and T.J.C. conducted the rat behavioral experiments and LHb innervation mapping. E.B.M., M.W.W., K.A.M., T.J.C., and K.A.M. performed the rat behavior data analysis. E.B.M. conducted the electrophysiology experiments and data analysis. M.M. supervised the MOR in situ hybridization study and analysis performed by D.J.B.; C.O., S.B., and D.J.B. conducted the VGluT2::Cre mouse fiber photometry experiments and subsequent data analysis. M.W.W., K.A.M., and E.B.M. wrote manuscript with contributions from T.J.C. and D.J.B. All authors read and approved the final version of the manuscript.

## Competing interests

The authors declare no competing interests.
