## [Peer Review File · Nature Communications]

A diencephalic circuit for opioid analgesia but not positive reinforcementREVIEWER COMMENTS

Reviewer #1 (Remarks to the Author):

This is an elegant series of preclinical studies examining mu opioid receptor (MOR) modulation of the projection from the later preoptic area (LPO) of the hypothalamus to the lateral habenula (LHb). This study begins with a behavioral examination of the effect of activating this receptor on the modulation of pain and drug-associated chamber preference. It is then established that MOR inhibits the glutamatergic inputs from the LPO as opposed to projections from other regions and possible GABAergic input from the LPO which also express MORs. Finally this study demonstrates that this pathway is activated by chronic pain and that MOR-mediated inhibition of the optogenetic activation of this pathway results in a conditioned place preference to the MOR-paired side in a painfree model.

The majority of the paper is well executed and the cellular experiments are thorough and concise with a clear description of MOR modulation of the LPO-LHb pathway, the strength of the paper and of interest to many in the field. However, parts of the behavioral experiments are unclear.

The authors consistently state that the LPO-LHb produces aversion relying on the literature for substantiation. However, the experiments and data presented in Fig 1 or 5 do not show aversion. Perhaps a different intervention such as carrageenan induced chamber preference is needed to demonstrate aversion as the control or this aspect of the paper needs to be refocused. The effect of DAMGO is then interpreted as a reduction in this aversive quality but these data show a simple DAMGO-paired chamber preference in keeping with known effects of MOR agonists. The difference between the effect of an opioid in reflexive vs affective tests of pain has been shown by others to be a dose-response effect specific to the type of test.

It is stated that both males and females mice show an effect of the MOR agonist in the LHb but the data shows that is only to an affective measure, DAMGO-induced CPP and a 10 fold higher dose of DAMGO is required. The dose used is very high (100uM) and controls for specificity should be included. In addition, the medial habenula is close and expresses the highest levels of MORs in the brain and mediate negative reinforcement. The discussion should be tempered accordingly.

The CPP data are condensed to a difference in score which may be confounded by the chamber design. Line 73 indicates a 3-chamber but the methods indicate a 2-chamber design.

SNI does not appear to result in a hyperalgesic state in male rats (Fig 1b), an expected effect of the model that was not included in the statistical analyses.

The use of the term 'negative reinforcement' as applied to an increase in DAMGO-paired CPP is confusing.

A schematic indicating the findings of the study in particular the pathway and effects of MOR activation on this pathway and cellular and behavioral outcomes would be beneficial.

What is a 'pseudorandom chamber assignment'. Surely assignment is either random or not.

The statistical analyses are appropriate.

The paper does need to be edited carefully. Numerous grammatical mistakes were noted, as shown below;

Line 25; "both MOR-agonist analgesia" and ?

Line 26; unclear

Line 99; we exemplified

Line 117; "is... during aversive states"- lacking verb participle

Line 153-154; comparator missing?

Reviewer #2 (Remarks to the Author):

Comments to the Authors:

In this very interesting study, Waung et al. used a multi-disciplinary and complementary approach including spared nerve injury (SNI) model of neuropathic pain, intra-LHb microinjections of MOR agonists, in vitro and in vivo optogenetics, fiber photometry calcium imaging, neuroanatomical tracing and in situ hybridization to elegantly demonstrate a critical role for a MOR sensitive glutamatergic pathway from forebrain to LHb in chronic pain. They showed that SNI-induced allodynia can be alleviated by intra-LHb MOR activation (more robustly in male rats with trending in females). More importantly, activation of MOR in the LHb of SNI rats (males and females) but

not control rats promoted conditioned place preference (CPP) suggesting that activation of MORs in the Lhb mediates analgesia without promoting positive reinforcement in pain-free subjects. Through a comprehensive set of electrophysiological experiments combined with in vitro optogenetics, they explored the sensitivity of five functional different synaptic inputs to the Lhb in rats and identified that LPO glutamatergic inputs to Lhb neurons were more robustly inhibited by MOR activation compared to other inputs. Their neuroanatomical tracing with in situ hybridization further substantiated high levels of MOR expression in LPO in comparison to other forebrain regions projecting to the Lhb. In addition, they also used anterograde tracing in addition to in vitro optogenetic studies to provide further evidence that ACC inputs to Lhb may be minimal with no functional connectivity between the two brain regions. Using SNI and Cre-dependent expression of GCaMP6m in VGlut2 Cre mice, they elegantly show the enhanced activity of LPO glutamatergic inputs to the Lhb in response to thermal stimulation in SNI mice compared to sham controls. To further validate the role of LPO –Lhb circuit in mediating aversive aspects of pain in rats, they then took a complementary approach of retrograde Cre delivery to express ChR2 in Lhb-projecting LPO neurons and activated these neurons during CPP to MOR agonists. They elegantly showed that optogenetic activation of this pathway in rats was indeed aversive and that MOR activation prevented the aversive conditioning of this pathway in rats. Overall, this study uncovers a novel role for this MOR sensitive neural circuitry in pain and opioid-induced analgesia. Overall, the experiments are appropriately designed, performed blindly and the data are clearly presented and conclusions are supported by the data. The paper should be of great interest to a broad audience, including those who study synaptic plasticity, Lhb circuits, pain, opioids and drug addiction. Therefore, the overall quality of the study is excellent and meets the scope and standard of this journal. There are only some clarifications/suggestions that need to be addressed.

- 1) There are no statistical analysis and interpretation for the effects of SNI and sex on paw withdrawal thresholds. As mentioned in method section by the authors, SNI reduces paw withdrawal thresholds and this allodynia is reversed by MOR agonists as shown in Fig 1b. The two way mixed ANOVAs could also be used here to show the effects of SNI itself and also sex in addition to the reported paired comparisons for DAMGO. It also seems that females have lower withdrawal thresholds in comparison to male rats. Is this basal gender-based difference reported before? The interpretations of these results can be included in the discussion which will be quite informative to the readers specifically those who are not familiar with these behavioral assays of neuropathic pain.
- 2) While the time course of study is depicted in Fig 1 a, it is still unclear how many days after SNI the allodynia testing was performed (15 days?). This should be clearly stated. Is there a time course study for SNI-induced mechanical allodynia that can be included similar to CFA-induced reduction in allodynia depicted in extended Fig 1 F? If so, this could also be very informative.
- 3) It is not surprising to this reviewer that optically-evoked PSCs as GABAA receptor-mediated IPSCs (outward currents recorded at -40mV) were as minimal or non-existent as shown in Extended Fig 5. Although the authors nicely verified the identity of these currents by DNQX and gabazine, and have isolated AMPAR-mediated EPSCs for DAMGO experiments, it is curious why they did not record IPSC currents at more positive holding potentials (like -20mV or even 0mV) with K-gluconate internals as others have shown. Again this is only a minor issue and a clarification for this would be appreciated.
- 4) Although MOR effects were studied on pharmacologically isolated optically-evoked EPSCs, this could not be achieved in the in vivo optogenetic study given that the expression of ChR2 for in vitro and in vivo optogenetics studies in rats was not confined to glutamatergic LPO neurons. Therefore, it would be useful to include more discussion about the limitation of optogenetic studies in rats and the possible effects of MOR activation on Lhb projecting GABAergic LPO neurons (inhibitory effects of MOR on GABA release in addition to glutamate release as reported by this group in their previous study) in addition to LPO-Lhb glutamatergic inputs and how MOR may have changed the balance of excitation to inhibition collectively at LPO-Lhb inputs in their study.
- 5) Degrees of freedom for F values of ANOVAs should be consistently reported throughout the manuscript as in Fig 5; $F(1,12)=5.439, \dots$

Reviewer #3 (Remarks to the Author):

This manuscript examines the role of mu opioid receptors (MORs) in the lateral habenula (LHb) in mediating the sensory and aversive effects of noxious input, particularly in the presence of neuropathic pain induced by the spared nerve injury (SNI) model. The studies use electrophysiology and optogenetic manipulation of specific glutamatergic circuits that project to the LHb to identify one projection from the lateral preoptic area (LPO) to LHb that is activated by noxious stimulation and sensitive to opioids. Optogenetic stimulation of this projection produces aversive behaviors that are inhibited by the MOR agonist DAMGO microinjections into LHb. Whole-cell patch-clamp experiments confirm that DAMGO inhibits LPO-LHb synaptic currents onto LHb neurons. The data show an interesting concentration-response shift between males and females where females are less sensitive to DAMGO. These exciting data are clearly presented in terms of the CPP behavior studies. The fact that opioids inhibit the LPO-LHb projection to produce negative reinforcement only in SNI-treated animals without CPP in naïve animals is an important and novel observation. Less convincing are the data showing that pain thresholds are altered by manipulation of LHb and the LPO-LHb circuit (see specific issues below). However, the overall conclusions of the manuscript are not dependent on there being an effect on pain thresholds. Increased rigor in the statistical design of those studies in particular could provide confidence in the interpretation that removal of “pain” results in the observed negative reinforcement observed by activating the LPO-LHb circuit. There are also some areas where increased information regarding experimental design would clarify displayed results.

Main comments:

1. Figure 1 – In panel b, it is not clear why the data are analyzed with a Wilcoxon signed rank test comparing control within a single treatment group with DAMGO microinjections. This study is interested in determining whether SNI decreases pain thresholds and if DAMGO microinjections reverse the decrease. Thus, this data should compare sham with SNI treatment between subjects with the within factor of vehicle/DAMGO. The variability in mechanical thresholds within each group suggests that there are no significant differences when analyzed this way. In addition, there appears to be no sham female controls. This data is integral to the interpretation of the results as the manuscript is written and needs to be analyzed appropriately.

In addition, it is concerning that icv DAMGO does not produce antinociception as it has been shown to be effective in previous studies, in addition to morphine and other opioids administered via that route.

2. Figure 2 – In panel a, it is not clear what the light blue shading is in that figure. Is this a single trace or averaged traces? What proportion of neurons had this response? How were these responses calculated? Are the data in panel b considered responders?

3. Figure 3 – In panel e, there is a weak return of the DAMGO inhibition by CTAP that appears to be consistent among experiments in the grouped data. Is the percent inhibition measured as DAMGO – control or is the reversal by antagonist included in that measure? This is important in determining the overall strength in opioid inhibition of the LPO-LHb projection.

4. Figure 5 – Panels a,b contain calcium imaging studies that show that there is an increase in LHb calcium levels that is time-locked to the paw withdrawal response to thermal stimulation. However, DAMGO microinjection into the LHb did not affect paw withdrawal responses to thermal stimulation. Does DAMGO inhibit the calcium response? Does stimulation of the LPO-LHb pathway induce increased calcium levels in LHb that are reduced by DAMGO? It is not clear what this data adds to the overall manuscript.

What is actually shown in panel b? Averaged calcium responses across animals? Or averaged traces from a single animal?

5. Discussion – page 11, lines 248-261 – the papers cited that LHb lesion relieves hyperalgesia in neuropathic pain models show partial reversal and the effects in these studies are not clearly presented.

Page 14, lines 311 – “Here we have identified a key circuit whose modulation relieves pain but does not generate reward in the absence of pain.” This section may have to be revised if revising

the statistical design does not support the change in % withdrawal threshold.

We thank the reviewers and editor for their careful consideration of our manuscript. We have made the editorial changes requested to meet the journal specifications and have created an open access online repository for our data on OSF. Please see our point-by-point response to the reviews below, reviewer comments in red.

Responses to Reviewer #1:

Reviewer #1 (Remarks to the Author):

This is an elegant series of preclinical studies examining mu opioid receptor (MOR) modulation of the projection from the later preoptic area (LPO) of the hypothalamus to the lateral habenula (IHb). This study begins with a behavioral examination of the effect of activating this receptor on the modulation of pain and drug-associated chamber preference. It is then established that MOR inhibits the glutamatergic inputs from the LPO as opposed to projections from other regions and possible GABAergic input from the LPO which also express MORs. Finally this study demonstrates that this pathway is activated by chronic pain and that MOR-mediated inhibition of the optogenetic activation of this pathway results in a conditioned place preference to the MOR-paired side in a painfree model.

The majority of the paper is well executed and the cellular experiments are thorough and concise with a clear description of MOR modulation of the LPO-IHb pathway, the strength of the paper and of interest to many in the field. However, parts of the behavioral experiments are unclear.

The authors consistently state that the LPO-LHb produces aversion relying on the literature for substantiation. However, the experiments and data presented in Fig 1 or 5 do not show aversion. Perhaps a different intervention such as carrageenan induced chamber preference is needed to demonstrate aversion as the control or this aspect of the paper needs to be refocused. The effect of DAMGO is then interpreted as a reduction in this aversive quality but these data show a simple DAMGO-paired chamber preference in keeping with known effects of MOR agonists. The difference between the effect of an opioid in reflexive vs affective tests of pain has been shown by others to be a dose-response effect specific to the type of test.

We agree with the reviewer that it strengthens the manuscript to directly show that in our animals optogenetic stimulation of the LPO inputs to the LHb produces aversion. To this end we have added a new experiment in the revision of the manuscript: by stereotaxically injecting CAV-Cre into the LHb and AAV2-hSyn-hChR2(H134R)-mCherry into the LPO in rats, we expressed ChR2 specifically in LPO neurons that project to the LHb. An optic fiber was implanted aimed at the LPO. In a conventional place conditioning assay, one environment was paired with stimulation of this pathway and one environment was paired with no stimulation. On test day, rats showed a significant CPA to the chamber in the apparatus paired with LPO-LHb stimulation (Figure 5c).

It is stated that both males and females mice show an effect of the MOR agonist in the IHb but the data shows that is only to an affective measure, DAMGO-induced CPP and a 10 fold higher dose of DAMGO is required. The dose used is very high (100uM) and controls for

specificity should be included. In addition, the medial habenula is close and expresses the highest levels of MORs in the brain and mediate negative reinforcement. The discussion should be tempered accordingly.

We agree with the reviewer and have revised the manuscript to more carefully and clearly describe the sex differences observed here (lines 120-131). Yes, the higher concentration DAMGO microinjection was 300 nL of 100 μ M DAMGO, or ~15 ng/side. We have also added information on sex differences in opioid antinociception to the discussion. In brief, females are less responsive to opioid induced antinociception across most, if not all, mouse and rat strains (lines 458-464).

The CPP data are condensed to a difference in score which may be confounded by the chamber design. Line 73 indicates a 3-chamber but the methods indicate a 2-chamber design.

The place conditioning boxes have 3 chambers; we have revised the language in the methods section to make it easier to read, "The conditioning boxes (Med. Associates, Georgia, VT, USA) have 3 divisions, two conditioning chambers (25 cm x 21 cm x 21 cm) with distinct visual (horizontal vs. vertical stripes) and textural (thick vs. thin mesh flooring) cues, separated by a third, smaller gray neutral chamber (12 cm x 21 cm x 21 cm)."

SNI does not appear to result in a hyperalgesic state in male rats (Fig 1b), an expected effect of the model that was not included in the statistical analyses.

We agree with the reviewer: we did expect to see more hyperalgesia in our mechanical threshold measurements in the male SNI rats. That said, a lack of hyperalgesia detection with von Frey measurements following SNI in male SD rats is not unheard-of (e.g. Alstrom et al., 2021). We did observe paw guarding in all included SNI rats, thus we interpreted these as successful inductions. We now mention this in the Methods section (lines 610-611). The experiments in question were designed to test whether MOR activation in the LHB generates antinociception, which is why our statistical testing was performed to specifically test this question. To respond to this inquiry we performed additional statistical comparisons to include here, mixed measures ANOVA where between animal "condition" = SNI/sham and within animal variable = saline/DAMGO, performed in R. P value for comparing thresholds between sham and SNI is highlighted in green. These comparison results should not be interpreted strongly since the data violate the requirements for parametric testing:

Male, intra-LHb DAMGO Von Frey thresholds:

ANOVA Table							
sts)	Effect	DFn	DFd	F	p	p<.05	ges
1	condition	1	14	0.150	0.704		0.009
2	drug	1	14	5.887	0.029	*	0.058
3	condition:drug	1	14	5.185	0.039	*	0.052

Bonferroni corrections									
drug	Effect	DFn	DFd	F	p	`p<.05`	ges	p.adj	
1 damgo	condition	1	14	0.22	0.646	""	0.015	1	
2 saline	condition	1	14	1.91	0.189	""	0.12	0.378	

Pairwise T test								
Condition.	group1	group2	n1	n2	p	p.signif	p.adj	p.adj.signif
1 sham-SNI	damgo	saline	8	8	0.963	ns	0.963	ns

2 sni damgo saline 8 8 0.0296 * 0.0296 *

Female, intra-LHb 10 μ M DAMGO Von Frey thresholds:

ANOVA

	Effect	DFn	DFd	F	p	p<.05	ges
1	condition	1	16	6.644	0.020	*	0.220
2	drug	1	16	1.522	0.235		0.030
3	condition:drug	1	16	0.162	0.693		0.003

Bonferroni

drug	Effect	DFn	DFd	F	p	`p<.05`	ges	p.adj
1	damgo condition	1	16	3.60	0.076	""	0.184	0.152
2	saline condition	1	16	5.55	0.032	***	0.258	0.064

t test

condition	group1	group2	n1	n2	p	p.signif	p.adj	p.adj.signif
1	sham-SNI damgo	saline	9	9	0.703	ns	0.703	ns
2	sni damgo	saline	9	9	0.223	ns	0.223	ns

Female, intra-LHb 100 μ M DAMGO Von Frey thresholds:

ANOVA

	Effect	DFn	DFd	F	p	p<.05	ges
1	condition	1	12	11.499	0.005	*	0.483000
2	drug	1	12	0.140	0.715		0.000307
3	condition:drug	1	12	1.892	0.194		0.004000

Bonferroni

drug	Effect	DFn	DFd	F	p	`p<.05`	ges	p.adj
1	damgo condition	1	12	10.5	0.007	*	0.467	0.014
2	saline condition	1	12	11.9	0.005	*	0.498	0.01

t test

condition	group1	group2	n1	n2	p	p.signif	p.adj	p.adj.signif
1	sham-SNI damgo	saline	6	6	0.92	ns	0.92	ns
2	sni damgo	saline	8	8	0.438	ns	0.438	ns

The use of the term 'negative reinforcement' as applied to an increase in DAMGO-paired CPP is confusing.

Classically, reinforcement of a behavioral response occurs with an outcome that is better than expected. Preference in a place conditioning assay is based on the association of a better than expected experience with specific spatial cues. In situations where there is no relevant aversive motivational state, the reinforcement is termed "positive". If the better experience is due to the reduction of an aversive motivational state, ongoing pain in the case of these experiments, it is termed "negative" reinforcement. We hope that this rephrasing here and in the manuscript (lines 48-50, 110-113) clarifies our meaning.

A schematic indicating the findings of the study in particular the pathway and effects of MOR activation on this pathway and cellular and behavioral outcomes would be beneficial.

We have added Figure 6, a schematic, in response to this suggestion (~ line 443).

What is a 'pseudorandom chamber assignment". Surely assignment is either random or not.

The subjects were assigned by alternating chambers after the first animal was assigned randomly. The correct wording for such a process is pseudorandom, as the subsequent assignments are determined by the assignment of the first animal, similar to a pseudorandom number generator.

The statistical analyses are appropriate.

The paper does need to be edited carefully. Numerous grammatical mistakes were noted, as shown below;

Line 25; "both MOR-agonist analgesia" and ?

Line 26; unclear

Line 99; we exemplified

Line 117; "is... during aversive states"- lacking verb participle

Line 153-154; comparator missing?

We thank the reviewer for the careful reading of the manuscript and we have made these and other corrections.

Responses to Reviewer #2:

Reviewer #2 (Remarks to the Author):

Comments to the Authors:

In this very interesting study, Waung et al. used a multi-disciplinary and complementary approach including spared nerve injury (SNI) model of neuropathic pain, intra-LHb microinjections of MOR agonists, in vitro and in vivo optogenetics, fiber photometry calcium imaging, neuroanatomical tracing and in situ hybridization to elegantly demonstrate a critical role for a MOR sensitive glutamatergic pathway from forebrain to LHb in chronic pain. They showed that SNI-induced allodynia can be alleviated by intra-LHb MOR activation (more robustly in male rats with trending in females). More importantly, activation of MOR in the LHb of SNI rats (males and females) but not control rats promoted conditioned place preference (CPP) suggesting that activation of MORs in the LHb mediates analgesia without promoting positive reinforcement in pain-free subjects. Through a comprehensive set of electrophysiological experiments combined with in vitro optogenetics, they explored the sensitivity of five functional different synaptic inputs to the LHb in rats and identified that LPO glutamatergic inputs to LHb neurons were more robustly inhibited by MOR activation compared to other inputs. Their neuroanatomical tracing with in situ hybridization further substantiated high levels of MOR expression in LPO in comparison to other forebrain regions projecting to the LHb. In addition, they also used anterograde tracing in addition to in vitro optogenetic studies to provide further evidence that ACC inputs to LHb may be minimal with no functional connectivity between the two brain regions. Using SNI and Cre-dependent expression of GCaMP6m in VGlut2 Cre mice, they elegantly show the enhanced activity of LPO glutamatergic inputs to the LHb in response to thermal stimulation in SNI

mice compared to sham controls. To further validate the role of LPO –LHb circuit in mediating aversive aspects of pain in rats, they then took a complementary approach of retrograde Cre delivery to express

ChR2 in LHb-projecting LPO neurons and activated these neurons during CPP to MOR agonists. They elegantly showed that optogenetic activation of this pathway in rats was indeed aversive and that MOR activation prevented the aversive conditioning of this pathway in rats. Overall, this study uncovers a novel role for this MOR sensitive neural circuitry in pain and opioid-induced analgesia. Overall, the experiments are appropriately designed, performed blindly and the data are clearly presented and conclusions are supported by the data. The paper should be of great interest to a broad audience, including those who study synaptic plasticity, LHb circuits, pain, opioids and drug addiction. Therefore, the overall quality of the study is excellent and meets the scope and standard of this journal. There are only some clarifications/suggestions that need to be addressed.

1) There are no statistical analysis and interpretation for the effects of SNI and sex on paw withdrawal thresholds. As mentioned in method section by the authors, SNI reduces paw withdrawal thresholds and this allodynia is reversed by MOR agonists as shown in Fig 1b. The two way mixed ANOVAs could also be used here to show the effects of SNI itself and also sex in addition to the reported paired comparisons for DAMGO. It also seems that females have lower withdrawal thresholds in comparison to male rats. Is this basal gender-based difference reported before? The interpretations of these results can be included in the discussion which will be quite informative to the readers specifically those who are not familiar with these behavioral assays of neuropathic pain.

We have added a sham SNI injury group for females (Figure 1). We also included the results of the mixed ANOVAs across the sham and SNI groups above in our responses to Reviewer 1. The means and ranges of 50% withdrawal thresholds for sham male vs female are very similar. We agree with the reviewer that we observed lower withdrawal thresholds in injured females compared to males. We did not test this outcome statistically because it was not the experimental question we posed in this study. That said, we agree with the reviewer that this sex difference is an important point in general, and we have expanded our description of the sex differences in the results section (lines 120-131) and included additional context from the literature on this point in the discussion (lines 458-464).

2) While the time course of study is depicted in Fig 1 a, it is still unclear how many days after SNI the allodynia testing was performed (15 days?). This should be clearly stated. Is there a time course study for SNI-induced mechanical allodynia that can be included similar to CFA-induced reduction in allodynia depicted in extended Fig 1 F? If so, this could also be very informative.

We performed sensory threshold testing on day 18 after injury induction. We have revised Fig. 1a to more clearly indicate each day in the experimental design. Studies consistently show that SNI in rodents produces persistent mechanical allodynia for at least 1 month and have been tested as far out as 30 weeks (Decosterd and Woolf, 2000). We added this information and relevant references to the text (lines 63-64).

3) It is not surprising to this reviewer that optically-evoked PSCs as GABA_A receptor-mediated IPSCs (outward currents recorded at -40mV) were as minimal or non-existent as shown in Extended Fig 5. Although the authors nicely verified the identity of these currents by DNQX and gabazine, and have isolated AMPAR-mediated EPSCs for DAMGO experiments, it is curious why they did not record IPSC currents at more positive holding potentials (like -20mV or even 0mV) with K-gluconate internals as others have shown. Again this is only a minor issue and a clarification for this would be appreciated.

When we initiated the experiments in Figure 4, inputs to the LHB from other brain regions, we had the same question the reviewer raises: what is a reasonable V_m at which to test for GABA_AR mediated IPSCs given our recording solutions? With the recording solutions used here, we previously found our experimental whole cell Cl^- reversal potential to be between -70 and -75 mV using GABA iontophoresis and IV analysis. We had also previously performed experiments investigating GABA_AR mediated synaptic input from the PAG to the VTA (Waung et al., 2019); in that study we had completed experiments with Cl^- reversal potentials used here or ~ 0 mV and voltage clamping the cells at -40 mV or -60 mV, respectively. We did not note an increased rate of apparent GABA_AR mediated connectivity in the recording configuration with the larger Cl^- driving force. This gave us confidence that the recording conditions utilized here were sufficient to detect GABA_AR mediated synaptic connections. Additionally, our experiments here investigating the glutamatergic and GABAergic inputs from other brain regions showed that holding at -40 mV was sufficient to clearly detect GABA_AR mediated synaptic connections, and these were often large in amplitude under these recording conditions (Fig 3c,f, extended data Fig 4c). In our experience with other datasets, averaging together 8 trials has been sufficient to reveal very small synaptic responses that were not readily detectable by eye in individual trials. This is one of the criteria we used to evaluate GABAergic connectivity in these experiments. In the case of our experiments here to test for local GABA_AR synapses within the LHB, the data inquired after here, the example data in Extended Figure 5b was the largest quantified deviation from baseline, and we now mention this in the legend. There is no discernable time locked synaptic event. Finally, for all of the experiments here testing for GABA_AR mediated synaptic connections we also considered that -40 mV a reasonable compromise between holding at a potential that would produce a large IPSC and a potential that is close to physiological for these neurons (in current clamp they often fire spontaneously or have a resting membrane potential -50 to -60 mV).

4) Although MOR effects were studied on pharmacologically isolated optically-evoked EPSCs, this could not be achieved in the in vivo optogenetic study given that the expression of ChR2 for in vitro and in vivo optogenetics studies in rats was not confined to glutamatergic LPO neurons. Therefore, it would be useful to include more discussion about the limitation of optogenetic studies in rats and the possible effects of MOR activation on LHB projecting GABAergic LPO neurons (inhibitory effects of MOR on GABA release in addition to glutamate release as reported by this group in their previous study) in addition to LPO-LHB glutamatergic inputs and how MOR may have changed the balance of excitation to inhibition collectively at LPO-LHB inputs in their study.

We agree with the reviewer on this point and have included a discussion of this point (lines 511-519). In this instance, we benefit from a projection that is heavily biased towards glutamatergic

neurons, which we demonstrate here electrophysiologically (Figure 3) and have demonstrated previously anatomically in the rat (Barker et al., 2017).

5) Degrees of freedom for F values of ANOVAs should be consistently reported throughout the manuscript as in Fig 5; $F(1,12)=5.439,....$

We have included the degrees of freedom with our statistical reporting in the figure legends as well as the Supplemental Table 1 that includes all the assumption testing results and the group comparisons.

Response to Reviewer #3:

Reviewer #3 (Remarks to the Author):

This manuscript examines the role of mu opioid receptors (MORs) in the lateral habenula (LHb) in mediating the sensory and aversive effects of noxious input, particularly in the presence of neuropathic pain induced by the spared nerve injury (SNI) model. The studies use electrophysiology and optogenetic manipulation of specific glutamatergic circuits that project to the LHb to identify one projection from the lateral preoptic area (LPO) to LHb that is activated by noxious stimulation and sensitive to opioids. Optogenetic stimulation of this projection produces aversive behaviors that are inhibited by the MOR agonist DAMGO microinjections into LHb. Whole-cell patch-clamp experiments confirm that DAMGO inhibits LPO-LHb synaptic currents onto LHb neurons. The data show an interesting concentration-response shift between males and females where females are less sensitive to DAMGO. These exciting data are clearly presented in terms of the CPP behavior studies. The fact that opioids inhibit the LPO-LHb projection to produce negative reinforcement only in SNI-treated animals without CPP in naïve animals is an important and novel observation. Less convincing are the data showing that pain thresholds are altered by manipulation of LHb and the LPO-LHb circuit (see specific issues below). However, the overall conclusions of the manuscript are not dependent on there being an effect on pain thresholds. Increased rigor in the statistical design of those studies in particular could provide confidence in the interpretation that removal of "pain" results in the observed negative reinforcement observed by activating the LPO-LHb circuit. There are also some areas where increased information regarding experimental design would clarify displayed results.

Main comments:

1. Figure 1 – In panel b, it is not clear why the data are analyzed with a Wilcoxon signed rank test comparing control within a single treatment group with DAMGO microinjections. This study is interested in determining whether SNI decreases pain thresholds and if DAMGO microinjections reverse the decrease. Thus, this data should compare sham with SNI treatment between subjects with the within factor of vehicle/DAMGO. The variability in mechanical thresholds within each group suggests that there are no significant differences

when analyzed this way. In addition, there appears to be no sham female controls. This data is integral to the interpretation of the results as the manuscript is written and needs to be analyzed appropriately.

For each dataset we evaluated the normality (Shapiro-Wilk Test) and homogeneity of variance (Levene's and Box tests), as well as noting if the dataset contains any outliers. This information is included in Supplementary Table 1. Details including the Statistic, p values, and related figure panel are listed there. Comparison tests were selected based on the assumption testing. As indicated in the table, we used the Wilcoxon signed rank test on the data in question because according to the Shapiro-Wilk Test of Normality we cannot assume they are sampled from a normally distributed population.

We agree with the reviewer regarding the value of the sham female control group and have added this group to Figure 1.

In addition, it is concerning that icv DAMGO does not produce antinociception as it has been shown to be effective in previous studies, in addition to morphine and other opioids administered via that route.

The goal of the i.c.v. experiments was to test whether we might be observing DAMGO induced antinociception due to the potential for non-specific spread of DAMGO in the ventricle, rather than a direct effect in the LHb. The dose of DAMGO that we microinjected i.c.v. is the same as the dose we injected into the LHb (10 μ M, 300 nL/hemisphere). In previous studies where the dose response relationship for i.c.v. DAMGO has been determined, the dose required for antinociception is greater than this amount (Al-Khrasani et al., 2007), consistent with our observations. The fact that this dose of DAMGO i.c.v. did not generate antinociception is consistent with our conclusion that the behavioral responses to DAMGO microinjections into the LHb are not confounded by any drug spread into the ventricles.

2. Figure 2 – In panel a, it is not clear what the light blue shading is in that figure. Is this a single trace or averaged traces? What proportion of neurons had this response? How were these responses calculated? Are the data in panel b considered responders?

In Figure 2a, this is the mean time course across all recorded neurons, \pm sem. We have included this information in the figure legend and put "n=15" in the figure panel. We did not specifically segregate responders vs non-responders in the previous version of the manuscript, and we included all recorded neurons in time course panels a (all neurons from SNI animals) and b, not just responsive neurons. We updated the *Data analysis for electrophysiology* section of the methods to describe how DAMGO effects were calculated and have added within cell analyses to classify responsive vs non-responsive neurons: for each individual neuron responses were quantified by comparing the mean values over the last 4 min of DAMGO application to the 4 minutes of baseline just prior to starting the DAMGO application, and individual responsive neurons were identified by comparing 30 sec binned samples during these two time windows with an unpaired t-test for that neuron. We encoded this information in the figure: filled circles indicate statistically identified responsive samples, and open circles indicate non-significantly responsive samples. We still include all of the recorded neurons from the SNI animals in the time course displays in Figure 2a,c.

3. Figure 3 – In panel e, there is a weak return of the DAMGO inhibition by CTAP that

appears to be consistent among experiments in the grouped data. Is the percent inhibition measured as DAMGO – control or is the reversal by antagonist included in that measure? This is important in determining the overall strength in opioid inhibition of the LPO-LHb projection.

All DAMGO effects are measured compared to baseline; reversal, or lack thereof, is not considered in the computation of the effect size. We have included additional details on this in the Methods section. The DAMGO effects in Figure 3d,e,f are reported as % of baseline evoked EPSC amplitude, comparing the mean of the amplitudes across last 4 min of data during the DAMGO application to the mean of the amplitudes to the 4 minutes just prior to starting the DAMGO application. We have updated the time course panel in Figure 3e to more clearly convey when we added CTAP and that CTAP was added in the presence of DAMGO. We agree with the reviewer that in the example traces in Figure 3e, left it is clear that we did not observe a full reversal of the DAMGO effect by CTAP, and this was common across neurons, as one can infer from the time course displayed in Fig 3e. We previously found that CTAP reversal of DAMGO inhibition at glutamatergic terminals in the LHb is quite slow and not complete (Margolis and Fields, 2016, Figure 2B), therefore the current observations are consistent with our previous observations. There are several biological reasons we might not achieve full DAMGO reversal with CTAP, such as that DAMGO drives internalization of the MORs providing less opportunity for CTAP, a large peptide that is unlikely to cross the plasma membrane, to access the receptor to reverse the agonist effects; or the signaling pathway that underlies the inhibition of neurotransmitter release in these terminals does not completely return to baseline when the MOR activation at the receptor protein ceases. Alternative antagonists such as naltrexone might reverse these effects more rapidly and/or completely, however we prefer to use CTAP since it is highly selective for MORs while naltrexone is also an antagonist at delta and kappa opioid receptors.

4. Figure 5 – Panels a,b contain calcium imaging studies that show that there is an increase in LHb calcium levels that is time-locked to the paw withdrawal response to thermal stimulation. However, DAMGO microinjection into the LHb did not affect paw withdrawal responses to thermal stimulation. Does DAMGO inhibit the calcium response? Does stimulation of the LPO-LHb pathway induce increased calcium levels in LHb that are reduced by DAMGO? It is not clear what this data adds to the overall manuscript.

In this revision, with optogenetic methods we have shown that stimulating the LPO projection to the LHb is aversive, that we can mitigate that aversiveness with MOR activation in the LPO, and that inhibiting this projection in animals with ongoing pain produces place preference. We believe the calcium imaging experiments in mice complement the causal optogenetic experiments by showing the neural responses to noxious stimulation are larger in the animals with ongoing pain. Moreover, the calcium imaging further serves to validate the role of specifically the glutamatergic component of the LPO -> LHb connection in pain-induced activity. Since the optogenetic experiments are more impactful, we have rearranged the order in which we present these datasets.

What is actually shown in panel b? Averaged calcium responses across animals? Or averaged traces from a single animal?

Panel b (now panel j) showed the average response across the entire group of mice. We have changed the figure caption to clarify this.

5. Discussion – page 11, lines 248-261 – the papers cited that Lhb lesion relieves hyperalgesia in neuropathic pain models show partial reversal and the effects in these studies are not clearly presented.

We thank the reviewer for pointing this out, and we have rewritten this paragraph (now starting at line 453).

Page 14, lines 311 – “Here we have identified a key circuit whose modulation relieves pain but does not generate reward in the absence of pain.” This section may have to be revised if revising the statistical design does not support the change in % withdrawal threshold.

We hope that the clarifications we have outlined above explain why we did not edit our interpretations of these data. We also reiterate here that our focus is on the relief of the subjective experience of ongoing pain rather than the sensory experience of the noxious stimulus.

REVIEWERS' COMMENTS

Reviewer #1 (Remarks to the Author):

The authors have addressed some of the raised concerns and clarified their findings. The paper has been strengthened and now presents a careful and exciting series of studies examining MOR modulation of the LPO-LHb pathway in chronic pain.

Reviewer #2 (Remarks to the Author):

The authors have done an outstanding job in this revision addressing all the issues raised in my initial evaluation of the manuscript and I have no further comments/suggestions.

Reviewer #3 (Remarks to the Author):

The revisions made in this manuscript adequately address the major concerns. The additional control group and demonstration that optogenetic stimulation of LPO neurons projecting to LHb induce conditioned place aversion, increases the overall evidence that the LPO-LHb circuit is involved with aversiveness of ongoing pain. Importantly, opioid-mediated inhibition of the circuit does not produce reward or conditioned place preference. The experimental design and interpretation of the results are solid with appropriate statistical design. The revision also includes additional information in methods and figure legends to clearly understand the experimental design. Overall, the results of this study are likely to be of interest to a wide audience and show an important distinction in the role of opioids in midbrain circuits.